# TESTAGENT: AN ADAPTIVE AND INTELLIGENT EXPERT FOR HUMAN ASSESSMENT

## ABSTRACT

Accurately assessing internal human states is critical for understanding their preferences, providing personalized services, and identifying challenges in various real-world applications. Originating from psychology, adaptive testing has become the mainstream method for human measurement. It customizes assessments by selecting the fewest necessary test questions (e.g., math problems) based on the examinee's performance (e.g., answer correctness), ensuring precise evaluation. However, current adaptive testing methods still face several challenges. The mechanized nature of most adaptive algorithms often leads to guessing behavior and difficulties in addressing open-ended questions. Additionally, subjective assessments suffer from noisy response data and coarse-grained test outputs, further limiting their effectiveness. To move closer to an ideal adaptive testing process, we propose **TestAgent**, a large language model (LLM)-empowered adaptive testing agent designed to enhance adaptive testing through interactive engagement. This marks the first application of LLMs in adaptive testing. To ensure effective assessments, TestAgent supports personalized question selection, captures examinees' response behavior and anomalies, and provides precise testing outcomes through dynamic, conversational interactions. Extensive experiments on psychological, educational, and lifestyle assessments demonstrate that our approach achieves more accurate human assessments with approximately 20% fewer test questions compared to state-of-the-art baselines. In actual tests, it received testers' favor in terms of speed, smoothness, and other two dimensions.

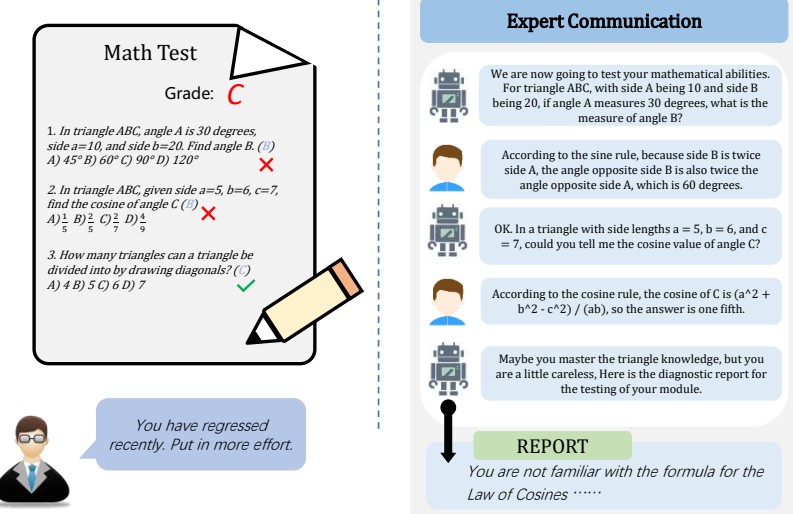

Figure 1: Examples of traditional testing: (a) Traditional paper-based tests where experts provide answers based on the test questions. (b) Our proposed testing expert system model. It will act as an expert, gradually assessing student abilities with as few interactions as possible.

# 1 INTRODUCTION

Designing effective assessments to evaluate specific human states is essential in various contexts, such as analyzing personality traits, diagnosing mental health issues, and measuring learning abilities (Kaufman et al., 2022; Laher et al., 2022). Traditional assessments often rely on paper-and-pencil formats, wherein questionnaires or selected questions are presented to participants. Based on participants' answer performance (e.g., answer correctness on math questions), experts can evaluate their states (e.g., mathematic ability). While these straightforward testing methods are functional, they are labor-intensive and require expert involvement. Moreover, the uniform testing environment can hinder personalization, complicating the tailoring of assessments to individual needs. Therefore, recent efforts focus on another testing form called computer-based Adaptive Testing (Liu et al., 2024), which aims to customizes assessments for each examinee by dynamically adjusting questions based on their performance. Adaptive testing allows for accurate and personalized evaluations with fewer questions, leading to widespread use in standardized testing, such as Graduate Management Admission Test (GMAT) and Graduate Record Examinations (GRE) (Mills & Steffen, 2000).

However, existing adaptive testing methods still face significant challenges, primarily due to three factors: (1) **Mechanized Testing Process.** Most adaptive algorithms are limited to fixed-answer questions, like multiple-choice formats. This rigidity can lead to guessing on unfamiliar questions (Brown, 2022), compromising assessment accuracy. Additionally, adaptive methods struggle with open-ended scenarios that involve varied answer formats, such as mathematical problem-solving. (2) **Noisy Answer Data.** In subjective assessments, answers may not reflect true internal states, introducing noise into the training of adaptive testing algorithms. For example, in personality tests, respondents may provide socially desirable answers rather than genuine feelings, leading to unreliable results (Stein & Swan, 2019). Similarly, in mental health evaluations, social pressures may cause individuals to conceal or misreport symptoms, skewing outcomes (McDonald, 2008). (3) **Coarse-Grained Test Output**. The adaptive question selection method provides a diagnosis value at the end of the test. It is difficult for the test-taker to make self-adjustments based on the diagnosis value. While traditional methods attempt to mitigate noise through expert-driven multi-step questions, the time and labor involved make large-scale implementation challenging (Josephson & Shapiro, 2013; Segal et al., 2019).

Recently, large language models (LLMs) have demonstrated impressive capabilities in human-like tasks, including reasoning, planning, and decision-making (Lee et al., 2024; Wang et al., 2024). This observation suggests the potential of LLM-driven agents to simulate human social behaviors across various contexts. Inspired by this potential, we propose the development of an LLM-based adaptive testing agent to overcome current testing limitations. Imagine that an intelligent agent, similar to a human expert, that can engage in interactive dialogues with examinees, analyze their responses, and dynamically generate personalized questions. Such an agent could transcend the mechanical constraints and noise-related issues inherent in traditional assessments.

Motivated by these considerations, we introduce **TestAgent**, an LLM-based agent designed to enhance adaptive testing through interactive engagement. This represents the first application of LLMs in adaptive testing. To ensure effective testing, TestAgent is designed to support **personalized question selection**, capture the examinee's **response behavior and anomalies**, and deliver **precise testing outcomes**. Specifically, TestAgent inherits the dynamic question selection capabilities of traditional adaptive testing, catering to personalized needs while improving testing efficiency. Additionally, an autonomous feedback mechanism and anomaly management module have been introduced to ensure a smoother and more intelligent testing process. TestAgent also generates detailed diagnosis reports to provide test-takers with a deeper understanding of their results, thereby making the testing experience more personalized and transparent, while significantly reducing resource costs. We conducted extensive experiments using datasets from three distinct domains, including personality measurement, educational math exam, and mental health test. The quantitative prediction results and qualitative analysis indicate that TestAgent's testing efficiency and methodology surpass traditional testing methods. Moreover, during actual tests, TestAgent was favored by testers for its speed, smoothness, and two other dimensions.

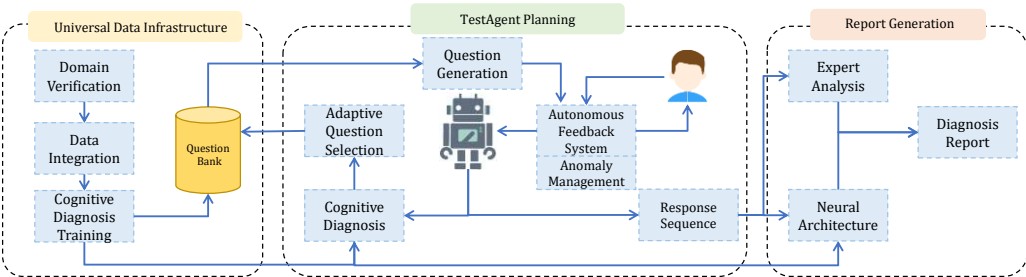

Figure 2: This is the overall framework of TestAgent. Universal The Data Infrastructure module is used to establish the question bank. The TestAgent Planning module outlines TestAgent's workflow. The Report Generation module is utilized to generate diagnosis reports.After the user answers question, the large language model summarizes the question and returns the labels to the cognitive diagnosis model. The cognitive diagnosis model assesses the current ability of the tester and uses a question selection algorithm to choose the best question from the question bank. Finally, the large language model communicates with the tester in a conversational manner.

## 2 TESTAGENT: GENERAL INTELLIGENCE TESTING EXPERT

### 2.1 PROBLEM DEFINITION

The goal of adaptive testing is to provide the test taker with tailored questions. It aims to do this in the fewest number of test rounds. It consists of two key components, the **Adaptive Question Selection** and **Cognitive Diagnosis**. After the test taker answers a question, Cognitive Diagnosis update their ability estimate based on the feedback of the question, and then further questions are selected based on the Adaptive Question Selection algorithm. The specific definition is as follows:

**Definition 1** (Definition of Adaptive Testing). *During the t-th step of testing, the test taker's response to question $q$ is $y$. The previous sequence of test question-answer pairs is denoted as $S = \{(q_1, y_1), \ldots, (q_t, y_t)\}$. At this point, the cognitive Diagnosis model updates the ability values based on S using cross-entropy loss. The question selection algorithm $\pi$ selects the best question based on the current $\theta_t$ for the test-taker to answer, i.e., $q_{t+1} \sim \pi(\theta_t)$. This process continues iteratively until it stops after T steps. The cognitive Diagnosis returns $\theta_T$ as the test result.*

There are several issues with traditional adaptive processes. First, label $y$ may not align with the true ability. In many cases, such as in math ability tests, the test-taker might randomly guess the correct answer which will significantly affect test accuracy. Second, test-takers may withhold information known about question $q$ due to various reasons, leading to inaccurate test results. Third, the cognitive diagnosis model outputs $\theta_T$ as the test result. However, this may not be intuitive for the test-taker. Test-takers tend to prefer receiving a diagnosis report that includes various analyses rather than a simple estimate of their abilities. These three issues will be addressed in our framework.

### 2.2 OVERVIEW

Similar to the process of adaptive testing, our framework also follows an iterative approach. The Figure 2 shows the pipeline of the entire working process of TestAgent. First, the Question Bank needs to be established. To do this, Domain Verification is required to determine the dimensions of the test and then followed by Data Integration. Cognitive Diagnosis Training will then complete the establishment of the Question Bank for use by TestAgent. Unlike traditional adaptive tests, our TestAgent transforms the entire testing process into a natural language conversation to break the Mechanized Testing Process at each step. As shown in Figure 1, instead of having the test-taker directly choose the answer $y$ for the question $q$, the TestAgent presents the question $q$ in the form of a natural language query posed by a character $C$. This is exactly what the Question Generation module does. Then the test-taker receives the transformed question $b = C(q)$ and responds with a conversation. Next, the TestAgent obtains $y$ from the conversation after passing through the **Autonomous Feedback Mechanism** and the **Anomaly Management** modules to address the issue of Noisy Answer Data. These two modules are aimed at obtaining more effective and stable labels.

Specifically, Autonomous Feedback Mechanism judges whether the label $y$ obtained by the agent is consistent with the response of the test-taker. If they are not consistent, the system automatically generates a similar question $b_{new}$ for the test-taker to answer, continuing this process until they are consistent. Anomaly Management is used to handle situations where the answer $y$ to a question exhibits anomalous behavior, such as when a tester tries to guess the answer or avoids responding to the question. If an anomaly occurs, it will use natural language guidance to progressively ask questions, reducing the likelihood of receiving misleading answers. After obtaining the accurate label $y$, the Cognitive Diagnosis module updates the test-taker's ability. Then Adaptive Question Selection module choose the most suitable question from the Question Bank. This process forms an iterative cycle.

To address the issue of Coarse-Grained Test Output, TestAgent utilizes Neural Architecture to provide initial analysis based on $\theta_T$ and the Response Sequence. This analysis is combined with Expert Analysis to ultimately form the Diagnosis Report. This report includes test results and suggestions for the test-taker. The implementation of these methods will be detailed in the following sections.

### 2.3 AUTONOMOUS FEEDBACK MECHANISM

During the conversational test, TestAgent analyzes the label $y$ based on the response from the test-taker. For some questions testing, TestAgent only needs to analyze whether the test-taker answered correctly like mathematical ability test. However, in more general tests like personality tests, TestAgent needs to analyze personality trait labels from a segment of daily dialogue of the test-takers. In such cases, it is highly likely that situations arise where the label cannot be analyzed. For example, if a test-taker responds with *"I don't know what to do"*, it clearly deviates from providing an answer and cannot be analyzed for a label. Therefore, we propose the Autonomous Feedback Mechanism to address this issue.

When the test-taker provides a response, the Autonomous Feedback Mechanism assesses from three perspectives: domain relevance, response alignment, and logical coherence to determine the outcome. From a domain relevance perspective, TestAgent leverage the intelligence of the Autonomous Feedback Mechanism associates questions with answers. If the response significantly deviates from the expected answer to the original question $q$, it is deemed unsuccessful. In terms of response alignment, responses are categorized into $M$ types representing the degree of alignment with the question. The value of $M$ is dependent on the specific test. For example, in a mathematical ability test, responses are either right or wrong. At this point, $M = 2$. However, for more complex tests like personality assessments, ranging from *"complete disagreement"* to *"complete agreement"* across seven dimensions, $M = 7$ making response analysis challenging. When response alignment is ambiguous, it is considered unsuccessful.

Regarding logical coherence, the Autonomous Feedback Mechanism evaluates whether the test-taker's response demonstrates internal logical consistency. Even if the response is related to the question, if it lacks coherence, contains contradictions, or is illogical, it is deemed unsuccessful. Logical coherence ensures that responses are not only superficially related to the question but also logically sound. If all three aspects are successful in their assessments, the label is returned; otherwise, based on the intelligence of the Autonomous Feedback Mechanism, a similar question is generated based on the properties of the question. This process continues until a label is determined.

### 2.4 ANOMALY MANAGEMENT

During specific tests, test-takers may guess the correct answer by chance, intentionally provide incorrect answers, or exhibit overconfidence in their responses, which can distort the assessment. Such anomalies can result in incorrect label $y$. These situations commonly occur in practice. The three most common types of anomalies in psychology are: Guessing Anomaly, Misleading Anomaly, and Overconfidence Anomaly. We are exploring these three types of anomalies.

**Guessing Anomaly** Test-takers may answer based on luck or incomplete understanding of the question, which does not accurately reflect their true abilities. In the case of Misleading Anomaly, test-takers deliberately provide incorrect answers, possibly due to lack of interest in the test or psychological resistance to the question. Detecting this anomaly is challenging. Overconfidence Anomaly occurs when test-takers demonstrate excessive confidence in their answers, even when uncertain.

While we cannot eliminate the subjective factors of test-takers entirely, we can strive to avoid these three types of anomalies to increase the accuracy of the tests. Anomaly Management, in conjunction with cognitive diagnosis, analyzes these anomalies. To assess the impact of anomalies on test results, Anomaly Management utilizes the cognitive diagnosis capability $\theta$ to assist in judgment. For a question $q$ where TestAgent receives feedback label $y$, cognitive diagnosis estimate the probability $P(q, y)$ of the test-taker answering question $q$ with label $y$ based on the current capability $\theta$. If the current capability value makes it difficult to answer with label $y$, Autonomous Feedback Mechanism is employed for judgment.

**Misleading Anomaly** Anomaly Management conducts reasoning based on context. By tracking the context of the test, inconsistencies or contradictions in a test-taker's responses to multiple questions on the same topic can be identified. For instance, if a test-taker provides a correct definition of a concept in one question but contradicts it in subsequent questions, it may be intentionally misleading. Anomaly Management then dissects the question to engage in more detailed dialogue with the test-taker to avoid such issues.

**Overconfidence Anomaly** Anomaly Management not only accepts the test-taker's response but also requests reasons or explanations for the answers. If a test-taker displays high confidence in their response but lacks sufficient reasoning or logic when explaining their choice, the model can determine their confidence is unfounded. Anomaly Management and Autonomous Feedback Mechanism complement each other, working in conjunction. Successful anomaly detection often requires a new round of questions to verify anomalies, increasing the precision of the tests.

## 2.5 TRAINING

**Cognitive Diagnosis Training** In the process of selecting intelligent questions, training the cognitive diagnosis model is one of the first challenges faced. We propose a general method for cognitive diagnosis training. We leverage the capabilities of GPT-4 to simulate examinees with different abilities. For instance, in MBTI tests, individuals can role-play different personalities to generate dialogue responses. Existing research has demonstrated that large models are reliable for simulating test-takers (Sekulić et al., 2024; Zhu et al., 2024). By facilitating continuous interaction between the model and all questions, response records are generated. Subsequently, the cognitive diagnosis model is trained based on the generated response records, specifically training the feature vectors $\beta$ for each question.

For different tests, the first step is to determine their test dimension $M$ called Domain Verification. For all interaction records $E$, the degree of answering questions is represented as $y \in [0, M]$, where the graded response model in Item Response Theory (IRT) can be applied. The probability of scoring less than m points can be calculated as the difference between the probability of scoring less than m points or more and the probability of scoring less than m + 1 points or more. For instance, $p_\theta(y = m|q) = p_\theta(y \geq m|q) - p_\theta(y \geq m + 1|q)$. Here: $p_\theta(y_i \geq m|q_i) = (1 + \exp(\theta - \beta_i^{(m)}))^{-1}$

These data are integrated to estimate the question features $F$. For example, question features can be computed as the proportion of correct answers. Additionally, data-driven techniques like cross-entropy loss can be employed to estimate these parameters. All question features are obtained by fitting response data: $\beta_i = \arg\min_\beta \sum_{e \in E} \sum_{i \in F} y_i \log p(y = y_i|q_i)$. Through this methodology, training of the cognitive diagnosis model can be achieved for existing tests.

**Diagnosis Report Generation** After conducting a certain number of test questions, cognitive diagnosis can analyze the abilities of the test-takers based on their responses. However, for personality tests like the MBTI, test-takers are more interested in receiving diagnosis reports. In this scenario, the vector $\theta$ is not interpretable. Therefore, generating diagnosis reports based on $\theta$ is crucial. To achieve this, we need to generate text labels for test-takers based on $\theta$ (for example, generating personality types in the MBTI test) and further generate diagnosis reports. Firstly, we train a classifier. This classifier can take $\theta$ as input and output the test results of the test-taker (for example, in the MBTI test, the classifier can determine the personality type based on $\theta$). During the question bank construction phase, we retained textual response records. By combining the test results, response records, and test reports provided by experts, we obtain fine-grained data for fine-tuning TestAgent to generate diagnosis reports. Once this fine-tuning is completed, we have finished the entire testing process. Test-takers can consider TestAgent as an expert in a certain field, engaging in multi-round dialogues to effectively assess their own skill levels and receive tailored recommendations.

## 3 EXPERIMENTS

### 3.1 EXPERIMENTAL SETTINGS

We used the proposed data synthesis method to annotate three datasets from different domains. These include the education dataset MATH, the personality measurement dataset MBTI, and the mental health test set SCL. The MATH dataset contains student practice logs related to math (A private data set). The MBTI dataset comprises questions from the 16-personality test. While the SCL-90 dataset includes questions from a depression tendency test. We fine-tuned the ChatGLM2-6B (GLM et al., 2024) series using comprehensive expert diagnosis reports and synthetic datasets as fine-tuning data. Training was conducted using the Lora method with a learning rate of 2e-5, all executed on Tesla A100:40G GPU.

### 3.2 ACCURACY TEST

**Data Partition and Evaluation Methods** To validate the efficiency of the adaptive testing method in selecting questions, a common practice involves randomly dividing each student's data into a query set $D_u$ and a support set $D_t$ (Ghosh & Lan, 2021). The support set $D_t$ is used to simulate the question selection process and estimate the final ability value $\theta_t$, while the query set $D_u$ is used to assess the accuracy of these estimates.

We performed 5-fold cross-validation on all datasets. For each fold, we allocated 60% of the students for training, 20% for validation, and 20% for testing. In each fold, we employed an early stopping strategy using the validation set to train the parameters for each method. To mitigate overfitting, we randomly shuffled these partitions at the beginning of each training epoch. The performance metrics for evaluation included Accuracy (Gao et al., 2021) and the Area Under the Receiver Operating Characteristic Curve (AUC) (Bradley, 1997).

**Compared Approaches** We employed three baselines for comparison: **Random**: This method randomly selects questions and serves as a reference for improvement compared to several baselines. **FSI**: (Lord, 2012): It utilizes maximum Fisher information to select questions. **KLI**: (Chang & Ying, 1996) It utilize Kullback-Leibler information to select questions. **MAAT**: (Bi et al., 2020) It employs an active learning (Krishnakumar, 2007) approach to measure question informativeness to select questions. The cognitive diagnosis model here adopts IRT (Ackerman et al., 2003) model.

**Result** Our TestAgent algorithm represents the new generation of adaptive testing, aimed at surpassing the limitations of traditional methods. In Table 1, we conducted a comprehensive comparison of the TestAgent with other model testing approaches. We not only provided accuracy (ACC) and area under the curve (AUC) metrics for test lengths of 5, 10, 20, and 50, but also used them as benchmarks to assess the performance of various models.

Our TestAgent framework demonstrates outstanding overall performance on these three datasets. Particularly noteworthy is the exceptional performance of the SCL dataset when utilizing the TestAgent model, showcasing the remarkable capabilities of TestAgent in handling complex datasets. Compared to traditional algorithms, our framework shows improvements in the majority of test steps, with the most significant enhancement seen at test step 5. On average, we achieved a relative improvement of 1.77% in AUC@5 and a notable increase of 0.91% in ACC@5. These results clearly demonstrate the highly accurate capability estimation provided by our framework.

### 3.3 SIMULATION OF ABILITY ESTIMATION

In adaptive testing evaluation, simulating the estimation of abilities is a fundamental evaluation technique (Vie et al., 2017). The purpose of testing is to accurately estimate the abilities of students. We conducted a simulation experiment on three datasets to estimate abilities. Specifically, we used the mean squared error $E[\|\theta_t - \theta_0\|^2]$ between the true ability of a test-taker $\theta_0$ and the ability at step t, $\theta_t$. Since the true ability $\theta_0$ is unknown, we approximated it by feedback from the test-taker answering all questions in the question bank (Bi et al., 2020; Cheng, 2009).

Figure 3 shows the metrics of different methods based on the IRT model on three datasets for testing questions ranging from 1 to 20 in total. As the number of selected questions increases, we find that the TestAgent method consistently achieves a lower estimation error. Compared to other algorithms,

Table 1: Prediction performance of different methods on ACC and AUC metrics for testee achievement prediction. The bold text indicates statistically significant superiority (p-value $\leq 0.01$) over the best baseline.

(a) Performances on MBTI

| Metric@Step | ACC@5 | ACC@10 | ACC@20 | ACC@50 | AUC@5 | AUC@10 | AUC@20 | AUC@50 |
|---|---|---|---|---|---|---|---|---|
| Random | 56.83±3.66 | 57.62±2.69 | 58.56±1.10 | 60.72±1.51 | 60.23± 1.61 | 60.92±1.53 | 61.65±1.57 | 64.04±1.98 |
| FSI | 58.70 ± 1.50 | 59.52±1.35 | 59.14±1.07 | 61.25±0.88 | 60.65±1.23 | 61.52±1.03 | 62.56±1.09 | 64.10±1.09 |
| KLI | 57.31±1.80 | 59.60±1.62 | 60.12±1.79 | 60.60±1.79 | 60.30±1.71 | 61.39±1.63 | 63.14±1.71 | 64.23±1.69 |
| MAAT | 59.60±1.95 | 59.68±1.84 | 59.89±1.89 | 60.45±1.99 | 61.91± 1.54 | 62.02±1.52 | 62.81±1.72 | 65.12±1.48 |
| TestAgent+FSI | 59.48±1.91 | **59.86±1.95** | **60.49±1.47** | 59.98± 2.13 | 61.60± 1.38 | 62.46±1.46 | **63.53±0.89** | 64.42±1.33 |
| TestAgent+KLI | 58.71±1.80 | 58.96±1.83 | 60.25±2.11 | **61.32±1.38** | 61.02±1.98 | 61.91±0.88 | 63.49±2.38 | **65.12±1.88** |
| TestAgent+MAAT | **60.21±2.04** | 59.48±2.34 | 60.24±2.21 | 61.31±1.40 | **62.11±1.49** | **62.75±1.61** | 63.21±1.71 | 64.88±1.82 |

(b) Performances on MATH

| Metric@Step | ACC@5 | ACC@10 | ACC@20 | ACC@50 | AUC@5 | AUC@10 | AUC@20 | AUC@50 |
|---|---|---|---|---|---|---|---|---|
| Random | 64.02±1.24 | 65.30±2.11 | 67.21±1.81 | 69.71±1.99 | 63.66±2.20 | 65.47±1.43 | 68.64±1.33 | 72.23±1.47 |
| FSI | 64.93±2.57 | 65.69±1.50 | 68.54±1.16 | 70.77±1.19 | 64.21±2.19 | 66.97±1.64 | 70.35±0.73 | 73.82±0.94 |
| KLI | 64.87±2.61 | 65.82±1.67 | 68.23±1.40 | 70.79±1.53 | 64.24±1.91 | 66.89±1.30 | 70.03±1.55 | 73.70±1.40 |
| MAAT | 64.45±2.12 | 65.71±1.79 | 67.92±1.70 | 70.23±1.78 | 64.09±0.95 | 66.34±1.01 | 69.40±1.66 | 73.23±1.60 |
| TestAgent+FSI | 65.32±1.67 | **66.28±2.25** | **69.39±1.41** | 71.02±1.81 | 64.84±0.14 | **67.87±1.80** | **70.91±0.94** | **74.00±1,23** |
| TestAgent+KLI | **65.52±0.92** | 66.19±1.70 | 68.97±1.59 | **71.20±1.91** | **64.90±2.06** | 67.38±1.90 | 70.84±1.98 | 73.97±1.84 |
| TestAgent+MAAT | 64.98±2.24 | 66.22±2.31 | 67.98±2.16 | 70.31±1.95 | 64.33±0.09 | 66.91±0.99 | 70.17±1.59 | 73.42±1.41 |

(c) Performances on SCL

| Metric@Step | ACC@5 | ACC@10 | ACC@20 | ACC@50 | AUC@5 | AUC@10 | AUC@20 | AUC@50 |
|---|---|---|---|---|---|---|---|---|
| Random | 54.74±1.31 | 55.45±1.95 | 57.97±2.20 | 62.82±2.16 | 48.17±2.25 | 49.59±1.07 | 54.94±1.82 | 63.89±1.68 |
| FSI | 60.00±0.51 | 62.12±1.61 | 64.44±1.24 | 66.76±1.44 | 58.04±0.59 | 62.85±2.41 | 67.08±1.19 | 69.16±1.19 |
| KLI | 60.50±1.56 | 63.73±0.98 | 64.74±1.51 | 65.95±1.46 | 64.82±0.88 | **68.02±1.55** | 69.49±1.40 |
| MAAT | 57.79±0.64 | 60.42±0.85 | 63.28±1.69 | 65.88±1.63 | 59.29±0.87 | 62.37±1.86 | 64.45±1.43 | 67.58±1.76 |
| TestAgent+FSI | **60.80±1.01** | 62.42±2.18 | 64.94±1.32 | **67.16±1.58** | 59.77±1.98 | 64.37±1.05 | 67.60±1.06 | 69.33±1.26 |
| TestAgent+KLI | 60.00±2.25 | **63.73±2.12** | **65.45±1.73** | 66.76±1.67 | **61.41±0.47** | **64.95±2.23** | 67.89±1.88 | **69.57±1.95** |
| TestAgent+MAAT | 58.28±2.16 | 61.13±1.82 | 63.48±2.15 | 66.37±1.86 | 60.02±2.84 | 62.88±2.24 | 64.92±1.45 | 68.30±1.69 |

TestAgent can achieve the same estimation error with fewer questions. It performs best on dataset SCL-90, reaching a similar level as others by step 15. On average, TestAgent can achieve the same estimation accuracy with 20% fewer questions, demonstrating its efficiency in estimating abilities, that is, reducing the length of the test.

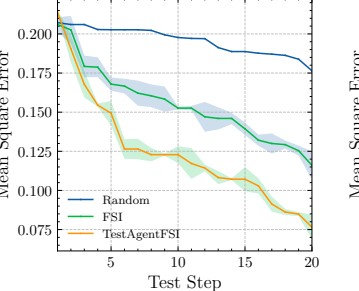 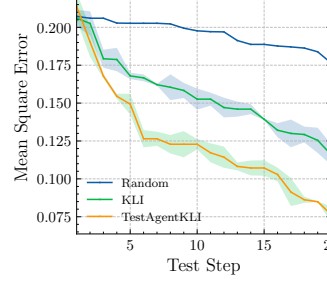 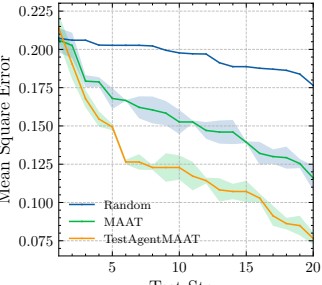

Figure 3: In dataset SCL, these three figures respectively show the Mean Square error of our method compared to traditional methods. It can be observed that in these three algorithms, incorporating the GPT module has led to an improvement of ability estimation errors.

## 3.4 EFFECTS OF LARGE LANGUAGE MODEL SIZE

To further explore the impact of model size on test accuracy, we conducted experiments using two different sizes of models, namely ChatGLM-6B and GLM-4-9B (GLM et al., 2024). The tests recorded the ACC accuracy at the fifth and twentieth steps, as shown in Figure 4. It can be observed that with the increase in model size, the accuracy continues to improve. This may be due to the enhanced analytical capabilities of larger models towards the labels, enabling them to approximate

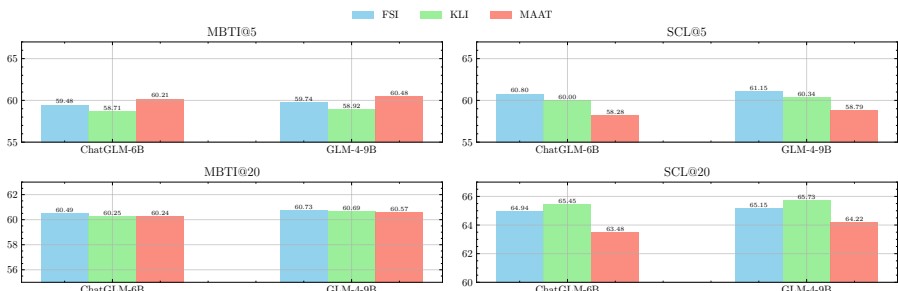

Figure 4: A comparative analysis of experiments conducted on the MBTI and SCL datasets using ChatGLM-6B and GLM-4-9B at steps 5 and 20.

real-world values more closely. Moreover, larger models exhibit stronger reasoning abilities and more pronounced anomalous responses, leading to more precise test results.

## 3.5 CASE STUDY

To better analyze the effectiveness of the TestAgent, we provide four case studies for examination, as shown in the Figure 5. The first case is the most standard scenario. The TestAgent asks a question, and the test-taker responds for assessment. The second case involves a situation where the label is ambiguous. The test-taker's response is difficult to interpret, so the process moves into the Autonomous Feedback Mechanism module. Utilizing the generative capabilities of large language models, similar questions are generated to resolve the uncertainty in label determination. The third case illustrates a situation where the test-taker is overly confident, leading to a testing error. The test-taker's response is too brief, so the TestAgent asks for further elaboration. This helps reduce hasty responses, enhances logical consistency, and increases testing stability. The fourth case demonstrates how guesses are handled. When the Anomaly Management module detects that the test-taker's response is likely a guess, similar questions are asked again. This reduces the impact of random guessing.

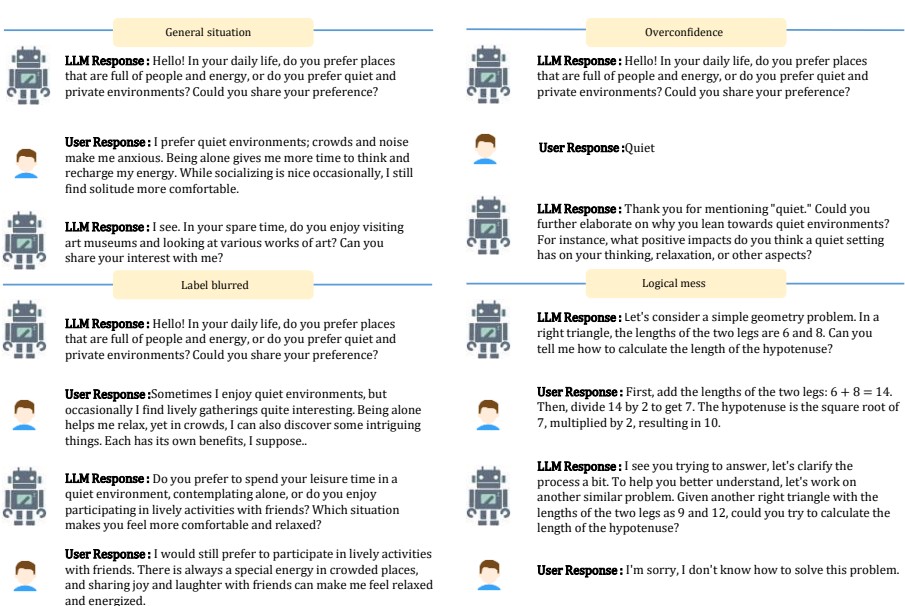

Figure 5: Examples showcasing case studies of TestAgent in different scenarios. It showcases the general case and the handling of anomalies in scenarios involving label blurred, overconfidence, logical mess.

Thus, the TestAgent processes natural language responses using text semantic analysis, summarizes these responses for the question selection algorithm, and then presents the next question, ensuring smooth interaction between the test-taker and the TestAgent. After several rounds of such interactions, the TestAgent generates a diagnosis report based on its engagement with the test-taker. A detailed example of this diagnosis report can be found in the appendix. Through this process, the TestAgent breaks the limitations of traditional testing methods.

## 3.6 QUALITATIVE ANALYSIS

TestAgent has powerful functionality. In order to better compare TestAgent's capabilities, we list several benchmarks for qualitative comparison. Computerized Adaptive Testing is a form of testing that adjusts the difficulty of questions based on the real-time performance of test takers, effectively improving test efficiency and accuracy. Multistage Testing is a staged test where each stage selects questions of different difficulty based on the test taker's previous performance. Interview is an interactive test form that evaluates the abilities, knowledge, and adaptability of test takers through face-to-face communication. Self-Assessment is a test form that allows test takers to assess themselves according to specific standards, emphasizing self-reflection and self-improvement. Simulation-based Assessment assesses test takers' performance and abilities in real-life situations through virtual scenarios or tasks. Table 2 shows the comparison. It can be seen that TestAgent has achieved in all aspects of evaluation.

| Benchmark | Low Cost? | Interaction Fluent? | No human Involvement? | High Time Efficiency? | Convenient to expand? | High Credibility? | High Engagement? |
|---|---|---|---|---|---|---|---|
| Paper-Pencil Test | ✓ | ✗ | ✓ | ✗ | ✓ | ✗ | ✗ |
| Computerized Adaptive Testing | ✓ | ✗ | ✓ | ✓ | ✓ | ✓ | ✗ |
| Mutistage Testing | ✓ | ✗ | ✗ | ✓ | ✗ | ✓ | ✗ |
| Interview | ✗ | ✓ | ✗ | ✓ | ✗ | ✓ | ✓ |
| Self-Assessment | ✓ | ✗ | ✓ | ✗ | ✓ | ✗ | ✗ |
| Simulation-Based Assessment | ✗ | ✓ | ✗ | ✓ | ✗ | ✓ | ✓ |
| **TestAgent** | ✓ | ✓ | ✓ | ✓ | ✓ | ✓ | ✓ |

Table 2: Comparison of Testing Methods Across Multiple Dimensions. The benchmark's testing methods may encounter in human daily tests. We design evaluation metrics to assess the functional correctness of test execution.

## 3.7 MULTIDIMENSIONAL EVALUATION

There are significant differences in design philosophy, execution, and user experience between traditional psychological tests and tests based on TestAgent. Evaluating which method is superior often varies due to personal preferences, testing purposes, and specific application scenarios. Therefore, we have adopted a more objective and comprehensive approach to assess the advantages of our innovative method. For this purpose, we carefully recruited 50 volunteers from different age groups, professional backgrounds, and educational levels to participate in this evaluation activity. These volunteers experienced the differences between our new method and the traditional MBTI testing method. Our goal is to conduct a comprehensive and detailed comparative evaluation of the two methods based on four core dimensions: "accuracy," "natural language fluency," "interaction experience," and "test speed." Volunteers were divided into two groups, each undergoing a different test first and then the other. After completing the tests, volunteers rated each dimension on a scale of 1 to 5 based on their experience.

Figure 6 displays the results, showing that the experience in natural language fluency, interaction experience, and test speed significantly surpassed traditional testing methods. This is because we conducted the tests entirely in a conversational format, enhancing user experience, and

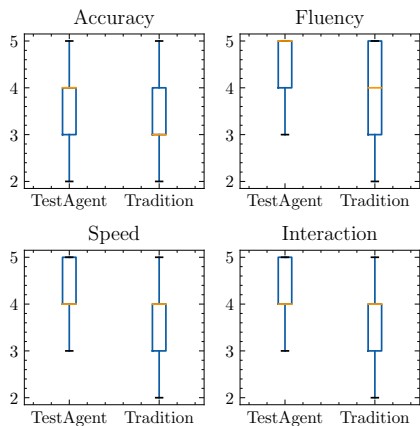

Figure 6: Displayed below are box plots comparing TestAgent with traditional testing across four dimensions.

combined with CAT technology to expedite the testing process. Through the real feedback and objective ratings of 50 volunteers, our new method has demonstrated advantages in "accuracy," "natural language fluency," "interaction experience," and "test speed." This outcome not only validates the feasibility and effectiveness of our innovative method but also provides new ideas and directions for the future development of the field of psychological testing.

### 3.8 FURTHER STUDY

**Challenges** In comparison to traditional testing, TestAgent is a conversational test based on a large language model. However, responses from large language models can exhibit fluctuations and even errors. Therefore, we conducted robustness testing to record instances of fluctuations and errors. We randomly sampled 50 interaction records in cases of failure, manually annotated and categorized their failure modes. The results are shown in Table 3: Primarily, there were some hallucination issues in the responses and summaries of answers (34%). Secondly, we also found instances of false negatives when using large model-based indicators, i.e., correct predictions that were misjudged as incorrect, but the proportion was relatively small (12%). In some cases, there were additional redundant conversational sentences generated in the summaries and responses to questions(26%). Additionally, at times, the model deviated from the role of the testing expert as specified in the prompts, assuming other identities for conversation, which is not in line with test guidelines(28%). These issues will be gradually addressed in future work.

Table 3: The error modes observed in random samples, the failure modes of TestAgent analyzed by humans, and their corresponding percentages.

| Error Type | Definition | % |
|---|---|---|
| Hallucination | Produce incorrect and nonexistent options | 34% |
| False Negative | Analysis of results with incorrect positive and negative analysis | 12% |
| Redundant answers | The question posed contains additional elements or deviates from the original question. | 26% |
| Inappropriate impersonation | Not testing according to the given role, saying things that do not match the test identity | 28% |

## 4 CONCLUSION

In this paper, we proposed an innovative conversational testing method that combined Large Language Models (LLMs) with Adaptive Testing technology, enhancing the flexibility and accuracy of traditional testing approaches. By introducing an LLM as a testing expert, we are able to dynamically adjust test content through multiple rounds of dialogues, thereby improving user experience and the precision of test results. Experimental results demonstrate that this method excels in assessments of psychology, abilities, and personality traits, effectively shortening testing time and enhancing the interpretability of diagnosis reports. In the future, we will introduce multimodal systems that utilize speech, images, and other modalities to assist large language models in testing can enhance the dimensions of testing. The TestAgent system, through its generated dialogues and personalized question selection, not only boosts testing efficiency but also offers fresh insights and directions for the future of psychological testing.

## IMPACT STATEMENT

In large language models combined with adaptive testing, different test takers may be recommended different questions, raising concerns about fairness. Our paper focuses on proposing a novel testing method, while fairness is another independent research area (Zhang et al., 2024), and thus is beyond the scope of our discussion.

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

# A RELATED WORK

Adaptive Testing typically includes two modules: the cognitive diagnosis model and the question selection algorithm. Below is an introduction to these two components.

**Adaptive Testing** (1) Cognitive Diagnosis Models. It is built on the foundation of psychometric theory, gaining popularity in assessments to provide more personalized feedback on students' latent abilities. It assumes that a test taker's ability remains constant throughout the testing process (Chang, 2015), allowing estimation of ability based on prior responses to questions using gradient-based optimization. The most classic form is the Item Response Theory (IRT) model (Ackerman et al., 2003). The simplest one-parameter logistic (1PL) model is represented as: $p$(correct response to question $j$) = sigmoid($\theta - b_j$), where $b_j \in \mathbb{R}$ represents the characteristics of each question, and $\theta \in \mathbb{R}$ is the student's ability to be estimated. Other representative models include Matrix Factorization (MF) (Koren et al., 2009; Toscher & Jahrer, 2010), Deterministic Inputs, Noisy-And gate (DINA) (De La Torre, 2009; Von Davier, 2014), and recently proposed Neural Cognitive Diagnosis Models (Cheng et al., 2019; Gao et al., 2022; Wang et al., 2023a; Shen et al., 2024) that leverage neural networks to model interactions between students and questions. In the case of specific CDM and response data, Maximum Likelihood Estimation (binary cross-entropy loss) is typically used to estimate $\theta$ for subsequent selection algorithm use.

(2) Selection Algorithms. The selection algorithm is a core component in achieving adaptivity in adaptive testing, aiming to estimate student abilities accurately with the fewest testing steps required. Traditional algorithms are based on uncertainty or information metrics such as the well-known Fisher information (FSI) (Lord, 2012)and other methods (Chang & Ying, 1996; Rudner, 2002; van der Linden, 1998; Veerkamp & Berger, 1997; Kang et al., 2017; Ma et al., 2023). In recent years, some data-driven methods have been proposed (Nurakhmetov, 2019; Zhuang et al., 2022; Ghosh & Lan, 2021; Wang et al., 2023b; Li et al., 2023; Yu et al., 2023)while some heuristic methods have also been proposed (Veldkamp & Verschoor, 2019; Gilavert & Freire, 2022; Feng et al., 2023; Mujtaba & Mahapatra, 2021; Yu et al., 2024) . However, most of these approaches are based on traditional paper tests, which lack the advantages of conducting assessments through a test booklet and may not achieve comprehensive testing.

**Large Language Model and AI Agents** In recent years, there have been many breakthroughs in various directions involving large language models. The emergence of agents based on large language models has garnered increasing attention from researchers as a burgeoning field. Numerous applications have been developed in specific domains and tasks, showcasing the powerful and versatile capabilities of these agents (Yao et al., 2022; Wang et al., 2022; Kim et al., 2024; Chan et al., 2023). Through domain fine-tuning, external knowledge bases, and more, a personal agent capable of assisting users in daily tasks can be created. With the enhancement of agent capabilities, human involvement becomes increasingly important to effectively guide and oversee the agents' actions, ensuring they align with human needs and objectives. Human-agent interaction agents can serve as guides for humans and have been applied in education (Kalvakurthi et al., 2023; Swan et al., 2023), health (Ali et al., 2020; Yang et al., 2024), and other fields (Gao et al., 2023; Schick et al., 2022), demonstrating the diverse capabilities of large language models. Large language models can also be used in a manner that establishes an equal partnership with humans, such as being empathetic communicators (Hasan et al., 2023; Liu-Thompkins et al., 2022) or functioning as human-level participants (Bakhtin et al., 2023; , FAIR).The measurement agent proposed in this paper is a universal measurement agent. By utilizing the corresponding dataset, one can obtain the corresponding agent using the method proposed in this paper, enhancing the effectiveness of human measurements across various domains and offering a novel measurement approach based on natural language dialogue in the testing field.

# B IMPLEMENTATION DETAILS

This section serves as supplementary details for the previous experiments.

## B.1 ABILITY CLASSIFIER TRAINING

Cognitive diagnostic models provide a vector $\theta$ as the diagnostic result; however, this is not interpretable. For the vector $\theta$, the MBTI test includes four dimensions: (I/E), (N/S), (T/F), and

(J/P). Therefore, we train a classifier where the input is the diagnostic model's $\theta$, and the output is a four-dimensional vector corresponding to these four dimensions, thus transforming the abstract diagnostic number into features. In specific terms, for a personality classification data labeled as $Y_{label}$, cognitive diagnostics provide a diagnostic result $\theta$ based on response to questions. Let $f$ be a mapping function that can map personality classifications to a 0-1 vector, for example, $f('ENFJ') = [1, 0, 1, 0]$. Let $g$ be the classifier we aim to train. The loss function can then be written as $L(\theta) = CrossEntropyLoss(g(\theta), f(Y_{label}))$. With this, the classifier training can be implemented.

### B.2 FINE-TUNE DETAILS

In this study, the fine-tuning process is based on the pre-trained ChatGLM model, aiming to customize the model for the specific personality diagnostic task to improve its performance in handling MBTI personality analysis tasks. To achieve this, we perform fine-tuning using LoRA (Low-Rank Adaptation) technology through the torchkeras framework.

**Data Processing:** The fine-tuning data is divided into three parts: instructions, character labels, and expert reports. The instruction is a simple prompt, formatted as follows: "Based on personality test classification and relevant dialogues, analyze the character traits and provide the corresponding diagnostic report."

Character labels include the labels obtained through the ability classifier training mentioned earlier.

Expert reports are the personality diagnostic reports provided by the official MBTI website for the 16 personality types.

Each piece of fine-tuning data consists of an input formed by combining the instruction and the character label, and the output is the diagnostic report suggestion, which corresponds to the expert's diagnostic report. Thus, the construction of fine-tuning data is completed.

**Parameter Settings:** In this work, several hyperparameters are carefully chosen for fine-tuning the model. The maximum sequence length is set to 1024 tokens, ensuring that input sequences longer than this are truncated.

For the Low-Rank Adaptation (LoRA) method, three key parameters are used: the rank $r$ is set to 8, which controls the size of the low-rank matrices; the scaling factor $\alpha$ is set to 32, which adjusts the influence of the low-rank adaptation; and the dropout rate $p$ is set to 0.1, which applies a 10% dropout during training to help with regularization.

Training-related hyperparameters include a batch size of 8, a learning rate of $2 \times 10^{-6}$, and a total of 10 training epochs. Additionally, early stopping is applied with a patience of 2 epochs, meaning that training stops if the validation loss does not improve over 3 consecutive epochs.

Finally, mixed precision training is employed with a setting of 'fp16' to improve computational efficiency, and when saving the model, the maximum shard size is set to 1GB, ensuring that the model is saved in manageable chunks for later use.

**Dataset Information**

Here we provide specific information for each dataset, along with concrete examples. The table displays the number of students, the number of questions, and the count of interaction responses for each dataset. Below are some specific question contents.

| DATASETS | Number of Testers | Number of Questions | Number of Questions |
|---|---|---|---|
| MBTI | 1000 | 60 | 60000 |
| SCL | 500 | 90 | 45000 |
| MATH | 1940 | 1485 | 61860 |

**MBTI**: *Your personal working style leans more towards spontaneous bursts of energy rather than systematic and sustained effort.*
**SCL**: *Feeling a decrease in energy and a slowing down of activities.*
**MATH**: *For a cylinder with a base radius of 1 and a height of 1, the surface area of the cylinder is.*

### B.3 MULTIDIMENSIONAL EVALUATION DETAILS

The multidimensional evaluation experiment involves 50 volunteers from different fields, who score on four dimensions: *accuracy*, *fluency*, *speed*, and *interaction*. **Accuracy** refers to how well the volunteer's results align with their actual situation and whether the final diagnostic recommendations are accurate. **Fluency** represents the smoothness of the test. **Speed** refers to the time taken to complete the test. **Interaction** measures the level of interactivity in the testing experience. However, human labeling can be subject to bias, which is inevitable. To reduce this bias, we have selected volunteers of varying gender, age, and educational background for the test.

| Type | Category | Percentage |
|---|---|---|
| **Gender** | Male | 62% |
| | Female | 38% |
| **Age** | 10-18 years old | 10% |
| | 18-30 years old | 46% |
| | 30-40 years old | 20% |
| | 40-60 years old | 18% |
| | 60-70 years old | 6% |
| **Education Level** | College degree | 46% |
| | No college degree | 54% |

Table 4: Demographic Information of Volunteers

We performed significance testing. We conducted hypothesis testing across different dimensions to eliminate bias in human annotations. The specific data is as follows:

#### GENDER: INDEPENDENT SAMPLES T-TEST

**Null Hypothesis** ($H_0$): There is no significant difference in the mean scores between males and females on a given dimension. That is, the mean scores of males and females are equal. **Alternative Hypothesis** ($H_1$): There is a significant difference in the mean scores between males and females on the given dimension. That is, the mean scores of males and females are different. Since the

| Dimension | t-statistic | p-value |
|---|---|---|
| Accuracy | -1.34 | 0.1805 |
| Fluency | -1.49 | 0.1372 |
| Speed | 1.05 | 0.2945 |
| Experience | 0.95 | 0.3416 |

Table 5: Independent Samples t-test for Gender

p-values are greater than 0.05, we cannot reject the null hypothesis.

#### AGE: ONE-WAY ANOVA

**Null Hypothesis** ($H_0$): There is no significant difference in the mean scores between the different age groups. That is, the scores of different age groups are similar. **Alternative Hypothesis** ($H_1$): At least one age group has a mean score that is different from the others. That is, there is a significant difference in scores between age groups. Since the p-values are greater than 0.05, we cannot reject

| Dimension | F-statistic | p-value |
|---|---|---|
| Accuracy | 1.0218 | 0.3971 |
| Fluency | 2.1243 | 0.079 |
| Speed | 2.0162 | 0.0936 |
| Experience | 1.3827 | 0.2413 |

Table 6: One-Way ANOVA for Age Groups

the null hypothesis.

EDUCATION LEVEL: INDEPENDENT SAMPLES T-TEST

**Null Hypothesis** ($H_0$): There is no significant difference in the mean scores between testers who have attended college and those who have not on a given dimension. **Alternative Hypothesis** ($H_1$): There is a significant difference in the mean scores between testers who have attended college and those who have not on the given dimension. Since the p-values are greater than 0.05, we cannot

| Dimension | t-statistic | p-value |
|---|---|---|
| Accuracy | -1.29 | 0.1984 |
| Fluency | 1.06 | 0.2867 |
| Speed | 1.26 | 0.2084 |
| Experience | -0.29 | 0.7653 |

Table 7: Independent Samples t-test for Education Level

reject the null hypothesis.

TEST COMPARISON: PAIRED T-TEST

We use the **paired t-test** to compare the score differences between traditional tests and TestAgent across each dimension. **Null Hypothesis** ($H_0$): There is no significant difference between traditional tests and TestAgent on a given dimension. **Alternative Hypothesis** ($H_1$): TestAgent outperforms traditional tests on the given dimension. Statistically, if the p-value is less than 0.05, the novel test on this dimension is considered significantly better than the traditional test. Since the p-

| Dimension | t-statistic | p-value |
|---|---|---|
| Accuracy | -2.56 | 0.01188 |
| Fluency | -6.53 | 2.80e-09 |
| Speed | -6.09 | 2.11e-08 |
| Experience | -6.46 | 3.87e-09 |

Table 8: Paired t-test for Traditional Test vs. TestAgent

values are all less than 0.05, we reject the null hypothesis and conclude that TestAgent outperforms traditional tests across all dimensions.

# C ADDITIONAL EXPERIMENTS AND ANALYSIS

## C.1 SIMULATION OF ABILITY ESTIMATION

In the main text, we only provided the test results of the SCL dataset. Here, we present the test results of two other datasets. The test results are as follows:

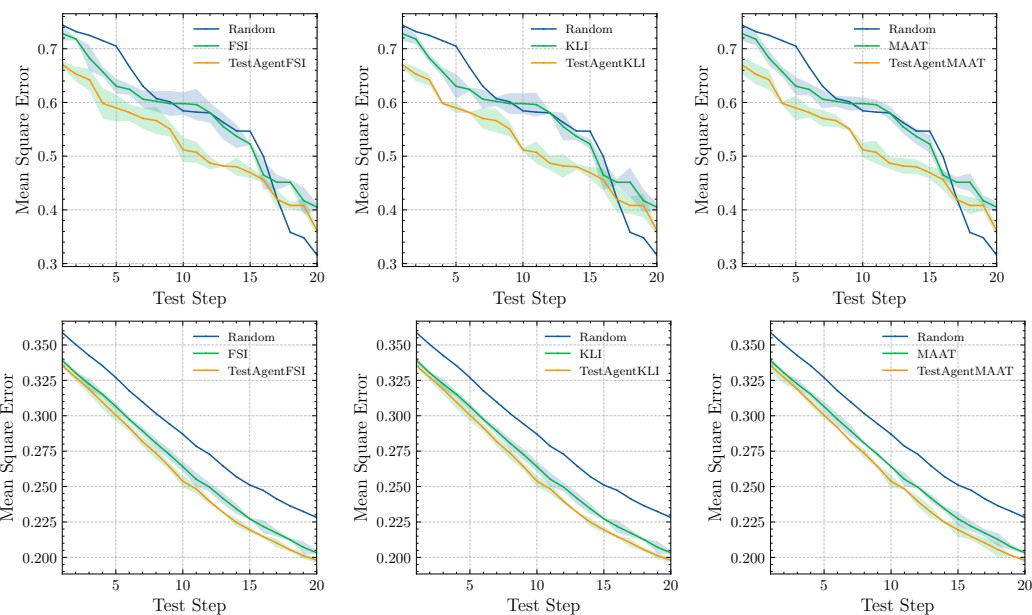

Figure 7: The three pictures above show the performance in the MBTI dataset, and below is the performance in the MATH dataset.

## D   DATA GENERATION ALGORITHM

---

**Algorithm 1** Data Generation

---

**Require:** Questions $Q$, GPT4 $G$, Test dimension $M$, Initialize parameters $\theta$, $\beta$

1: **for** each Epoch **do**
    The large language model $G$ plays different roles to answer questions $Q$, generating responses $Y$ Combine the answers with the questions to obtain the data $\{(q_1, y_1), \ldots, (q_n, y_n)\}$ where $q_i \in Q$ and $y_i \in [0, M]$
2: **end for**
3: **while** not converged **do**
4:    Randomly sample a mini-batch of students with training set $\Gamma$ and validation set $\Omega$
5:    Train using the training set $\Gamma$ and loss function $L(q, y; \theta)$, where $L(q, y; \theta) = \mathbb{E}_{y \sim p(y|q)}[-\log p_\theta(y|q)]$
6:    Validate using the validation set $\Omega$; stop training when converged
7: **end while**
8: Obtain the complete labeled data $\{(q_1, \beta_1), \ldots, (q_n, \beta_n)\}$

---

## E   PROMPT

This includes the segments mentioned in the main text. These segments include tag judgment, Auto Feedback Mechanism, Anomaly management, problem transformation, and other methods. The table below specifically displays the inputs and prompt of each method

## F   EXAMPLES

This section provides some examples of failures during testing and offers a sample diagnosis report.

Table 9: An example prompt of Auto Feedkback Mechanism

| | |
|---|---|
| Input | User Response,Question |
| Situation | Auto Feedkback Mechanism |
| Prompt | You will receive two inputs: the test-taker's response and the current question. Your task is to evaluate whether the response is relevant, logically sound, and easy to judge. Return the following results: - If the response is completely unrelated to the question, return 'False'. - If the response is difficult to judge, such as the test-taker saying, 'I don't know what to answer,' return 'False'. - If the response is logically inconsistent, also return 'False'. Else return 'True' If you return False, generate a new question that is similar to the original question but potentially easier or more specific. Otherwise, proceed with further analysis and provide appropriate feedback." Examples: 1. Unrelated responses (Return 'False'): - Question: '"What is Newton's third law?"' - Response: '"I like eating pizza."' 2. Difficult-to-judge responses (Return 'False'): - Question: '"Explain the process of cell division."' - Response: '"I don't know how to explain it."' 3. Logically inconsistent responses (Return 'False'): - Question: '"How do you prove a triangle is equilateral?"' - Response: '"Because it has three angles, it must be equilateral."' - Question: '"What is the relationship between current and voltage?"' Question Generation: Original Question: "Do you prefer being in a lively environment or being alone?" Generated Similar Question: "Do you enjoy socializing with others or spending time by yourself?" Question :[Question] Response:[User Response] |

Table 10: An example prompt of Anomaly management

| | |
|---|---|
| Input | User Response,Question |
| Situation | Anomaly management |
| Prompt | Task: If it is detected that the respondent is unwilling or reluctant to answer, break the original question into smaller, easier-to-answer questions, and gradually guide the respondent to provide more information. Else return 'True' to go next stage. If the respondent's answer is too brief, provide a short prompt to encourage further elaboration. Example 1: Avoiding the question Current question: "Do you prefer spending time alone or socializing with others?" Respondent: "Well, it depends." Guidance: "Could you share a specific example? For instance, when you're working, do you prefer working alone or collaborating with a team?" Example 2: Answer is too brief Current question: "When making decisions, do you rely more on logic or intuition?" Respondent: "Logic." Guidance: "Could you elaborate? In what situations do you tend to rely more on logic rather than intuition?" Question:[Question]. User Response: [User Response] |

## F.1  Failure

## F.2  Diagnosis Report

Table 11: How to play a role and get the prompt for generating data

| Input | Question Bank , The simulated role |
|---|---|
| Situation | Question Response Generation |
| Prompt | Please act as a [role] and respond to each question using the following rating scale. Your response should reflect your attitude or opinion towards the question, using the rating scale to indicate your answer: |
| | 0: Completely Disagree 1: Strongly Disagree 2: Mildly Disagree 3: Neutral 4: Mildly Agree 5: Strongly Agree 6: Completely Agree Requirements: |
| | Understand the Question: Carefully read each question and provide a response based on your understanding and hypothetical background as [role]. |
| | Select an Appropriate Rating: Choose the most appropriate rating (from 0 to 6) based on the content of the question. Example: |
| | Question: Do you believe that teamwork is more effective than working alone in urgent situations? |
| | Response: 5 (Strongly Agree) — In urgent situations, teamwork brings together more skills and resources, which helps to resolve issues more quickly. Question: Do you think frequent communication at work reduces productivity? |
| | Response: 1 (Strongly Disagree) — Although leaders should consider team members' opinions, final decisions should be based on overall interests and goals. Question: Do you believe that employee autonomy fosters innovation within a company? |
| | Response: 6 (Completely Agree) — Providing employees with autonomy can stimulate creativity and innovative thinking, contributing to the development of new solutions and products. Question List: |
| | Please provide your ratings and brief explanations for each question based on the role of [role]. Give me an answer. The format is as follows: 'Question 1': 'Answer': 0, 'Response': 'I feel very tired from making new friends, so I don't want to make new friends'. |

Table 12: Transforming Rigid Questions

| Input | The rigid question selected from the question bank |
|---|---|
| Situation | After selecting the questions, LLM transforms them |
| Prompt | You are an expert in conversation generation, specializing in transforming mechanical questions into lively, natural dialogue forms. Your task is to make these questions more attractive and interactive to spark the interest and positive response of the other party. Please refer to the following examples and transform each mechanical question into a natural conversational style.Mechanical Question: "Do you like visiting art museums?" Natural Dialogue Form: "Hi! In your leisure time, do you choose to visit art museums to appreciate various artworks? Or do you have any particular exhibitions or artists that you especially like?" |
| | Mechanical Question: "Do you enjoy teamwork?" Natural Dialogue Form: "Hello! When you are at work, do you find it more enjoyable to collaborate with a team? Or do you prefer completing tasks on your own? I'm curious to know what specific appeal or challenges teamwork holds for you." |
| | Mechanical Question: "Do you like traveling?" Natural Dialogue Form: "Hey! If given the opportunity, where do you most enjoy traveling to? Is there a place that has left a lasting impression on you, or experiences during your travels that excitep you the most?" |
| | Ensure the tone of the conversation is friendly and engaging. Make the questions interactive to encourage sharing more details. Use a casual, natural language to make the conversation more approachable. Please follow these guidelines to transform each mechanical question into a natural, lively conversation form to facilitate pleasant communication. Only return the natural dialogue form. Mechanical Question:[Do you like dog] , Output: |

Table 13: The prompt of summarizing the tester's response.

| | |
|---|---|
| Input | The tester's response." |
| Situation | The summary by LLM after the tester's response. |
| Prompt | You are a professional psychological test analyst, tasked with analyzing the degree of agreement of the respondents to each question based on their answers. The scoring ranges from 0 to 6, where 0 stands for "completely disagree" and 6 stands for "completely agree".
Scoring Guide:
0: Completely Disagree 1: Strongly Disagree 2: Mildly Disagree 3: Neutral 4: Mildly Agree 5: Strongly Agree 6: Completely Agree
Please provide the corresponding agreement ratings and brief explanations based on the respondents' answers below:
Example Questions and Responses:
Question: "You prefer lively places over being by yourself." Respondent's answer: "Yes, being alone is too boring." Rating: 6 Explanation: The respondent expresses a strong preference for lively environments. The answer clearly indicates their liking for lively places with almost no hesitation or reservation.
Question: "You feel more motivated when working in a team." Respondent's answer: "I think teamwork can be more effective at times, but not always." Rating: 3 Explanation: The respondent's answer shows a neutral stance. While acknowledging the effectiveness of teamwork at times, there is no explicit strong preference or opposition.
Question: "You tend to rely on logic rather than emotions when making decisions." Respondent's answer: "I usually consider facts and logic, but emotions can also influence me at times." Rating: 4 Explanation: The respondent tends to rely on logic but also acknowledges the influence of emotions on decision-making, hence the rating falls within the mildly agree range.
Ensure that your ratings and explanations accurately reflect the degree of agreement for each answer and provide specific reasons to support your ratings. |

Table 14: Example of Hallucination

| | |
|---|---|
| Error Type | Hallucination |
| Question | Do you often make new friends? |
| Example | Response: -1 (Completely Disagree) — I generally find social interactions draining and prefer spending time alone or with a small group of close friends. |
| Example | As someone who is more introverted, I find social interactions can be quite exhausting. While I value meaningful connections and enjoy spending time with close friends, I often prefer solitude or low-key social settings over large gatherings or frequent social activities. |

Table 15: Example of False Nagetive

| Error Type | False Nagetive |
|---|---|
| Question | You're definitely not the artistic type. |
| Example | Response: 6 (Completely Agree) — I really enjoy solitary activities like art creation. I appreciate art pieces and engage in art-related activities. I also like sharing these experiences with others. |

Table 16: Example of redundant answers

| Error Type | Redundant answers |
|---|---|
| Question | Even a small mistake can make you doubt your overall abilities and level of knowledge. |
| Example | Response: None (Error) : The responder acknowledges a lack of confidence and mentions that even small mistakes make them doubt their abilities and level of knowledge, causing significant distress. |

Table 17: Example of inappropriate impersonation

| Error Type | Inappropriate impersonation |
|---|---|
| Question | You avoid taking on a leadership role in group settings. |
| Example | TestAgent:"LLM Response to Tester: Hi! When you're in a group setting, do you find yourself stepping back and letting others take the lead, or do you sometimes feel inspired to jump in and guide the team? I'm curious—what's your take on being in a leadership role? Is there something specific that draws you to it or makes you shy away from it?" |

Table 18: A example of an MBTI test diagnosis report.

| Type of Test | MBTI |
| --- | --- |
| Diagnosis Report | Diagnosis Report for Alice |
| | Name: Alice Gender: Female Age: 18 years old MBTI Type: INTJ (Introversion, Intuition, Thinking, Judging) |
| | Personality Type Overview INTJ Overview: |
| | Alice's MBTI type is INTJ, often referred to as the "Architect" or "Mastermind." INTJs are known for their strategic and analytical mindset, which is reflected in their preference for Intuition (N) over Sensing (S), and Thinking (T) over Feeling (F). This personality type is characterized by the following traits: |
| | Introversion (I): Alice is likely to be reserved and prefers spending time alone or with a small group of close friends. She recharges by reflecting internally rather than engaging in large social gatherings. Intuition (N): Alice focuses on the big picture and future possibilities rather than immediate details. She tends to think abstractly and is interested in exploring complex concepts and ideas. Thinking (T): Alice values logic and objectivity. She makes decisions based on rational analysis and objective criteria rather than personal feelings. Judging (J): Alice prefers structure and organization. She likes to plan ahead and is often decisive, seeking closure rather than leaving things open-ended. Main Characteristics: |
| | Strategic Thinking: Alice is adept at seeing long-term implications and developing comprehensive plans to achieve her goals. Independence: She values her autonomy and often works best on her own or in leadership roles where she can direct her own efforts. High Standards: Alice has a strong drive for excellence and tends to set high standards for herself and others. Reserved: She may come across as reserved or distant, particularly in unfamiliar social settings. Career and Job Advice Career Development Advice: |
| | Leadership Roles: Alice's strategic mindset and ability to see the big picture make her well-suited for leadership positions. Careers in management, strategic planning, or executive roles could be fulfilling. Analytical Fields: Fields that require complex problem-solving and strategic thinking, such as engineering, finance, or research, may align well with Alice's strengths. Autonomy and Innovation: Jobs that offer independence and opportunities to innovate will cater to Alice's preference for working alone and developing new solutions. Career Growth: Alice should seek roles that allow her to work on long-term projects and provide opportunities for personal growth and development. Influence on Job Performance: |
| | Alice's ability to strategize and plan effectively can lead to high job performance, especially in roles that value long-term vision and critical thinking. Her high standards might lead to perfectionism; thus, it's important for her to balance her expectations with practical constraints. Career Satisfaction: |
| | Alice will likely find satisfaction in roles that challenge her intellectually and offer opportunities for advancement. She may need to ensure she has sufficient time for personal reflection and avoid burnout from overcommitment. Interpersonal Relationship Advice Strengths: |
| | Insightful: Alice's ability to analyze situations and understand complex dynamics can be beneficial in both personal and professional relationships. Reliable: Her structured approach and high standards can make her a dependable partner or colleague. Challenges: |
| | Communication: Alice's reserved nature and focus on logic may sometimes make it difficult for her to connect emotionally with others. She might need to work on expressing her feelings and being more open. Perfectionism: Her high standards might lead to frustration if others do not meet her expectations or if she feels things are not progressing as planned. Improvement Suggestions: |
| | Active Listening: Alice should practice active listening to better understand others' perspectives and build stronger connections. Empathy: Developing empathy and showing appreciation for others' feelings and contributions can improve her relationships. Personal Growth Advice Leveraging Strengths: |
| | Goal Setting: Alice should continue setting clear, long-term goals and devising strategic plans to achieve them. Learning Opportunities: Pursuing continuous learning and self-improvement will keep her intellectually stimulated and satisfied. Areas for Development: |
| | Emotional Intelligence: Alice could benefit from enhancing her emotional intelligence, including understanding and managing her own emotions and those of others. Flexibility: While structure is valuable, being open to adapting her plans and expectations can help Alice navigate unforeseen challenges and foster better collaboration. Common Misconceptions Misconceptions to Clarify: |
| | Misconception: INTJs are often seen as cold or distant. |
| | Clarification: While Alice may appear reserved, this doesn't mean she lacks warmth or compassion. It's more about her preference for processing emotions internally. Misconception: INTJs are rigid and inflexible. |
| | Clarification: Although Alice values structure, she is also capable of adapting her plans when necessary, especially if it aligns with her strategic goals. Misconception: INTJs are uninterested in others' opinions. |
| | Clarification: While Alice values logical analysis, she can still be open to feedback and differing perspectives if they contribute to her understanding of a situation. |

