# OpenReview forum: "TestAgent: An Adaptive and Intelligent Expert for Human Assessment"
_ICLR.cc/2025/Conference — ICLR 2025 Conference Withdrawn Submission_

### Official Review · Reviewer_mcjy · 2024-10-22

[review text omitted: it was posted to a different submission]

---

> ### Author Response · Authors · 2024-11-20
>
> Thank you for your affirmation and feedback on our article. It seems that there might have been a mix-up in your review comments between our paper and paper ID 6485. Could you please synchronize the feedback for us? Here, we provide a detailed response to the review comments mentioned in the review of paper ID 6485.
>
> > **Q1:** In the final step of the framework, expert analysis is still required. This feels a bit contradictory to the goal of improving automation. Since you're already using LLMs, why not go a step further and design an evaluation agent to replace or supplement the expert analysis? This could boost efficiency and reduce dependence on experts.
>
> We use expert analysis here to reduce diagnosis hallucinations of large language model and enhance the interpretability of diagnosis reports. If a agent is designed to replace diagnosis reports without any fine-tuning, it cannot guarantee the professionalism of the diagnosis reports generated by this LLM. Similarly, using an agent as an expert analysis can sometimes result in very neutral statements, making it difficult for testers to obtain useful information from such reports. So it is important to employ expert analysis supplemented with fine-tuning to make the results more convincing and reduce hallucinations of large language models.
>
> > **Q2:** While the framework diagram is clear, I think it could be made even more detailed.
>
> We provided a specific example. Below is the complete process of implementing the MBTI test to facilitate better understanding.
>
> **Universal Data Infrastructure**
>
> | Testing Step | Module Name                  | Specific content                                             |
> | ------------ | ---------------------------- | ------------------------------------------------------------ |
> | Step 1       | Domain Verification          | MBTI test requires 4 dimensions: (I/E), (N/S), (T/F), (J/P)  |
> | Step 2       | Data Integration             | Generate MBTI data using GPT-4 to simulate interaction data  |
> | Step 3       | Cognitive Diagnosis Training | Train a cognitive diagnosis model using the simulated interaction data |
>
> **TestAgent Planning**
>
> | Testing Step | Module Name                                       | Specific content                                             |
> | ------------ | ------------------------------------------------- | ------------------------------------------------------------ |
> | Step 1       | Question Generation                               | Generate questions in a conversational format from the question bank for the tester |
> | Step 2       |                                                   | Tester responds to the questions                             |
> | Step 3       | Autonomous Feedback System and Anomaly Management | Analyze responses, and if anomalies in labels are detected, generate new questions and return to Step 2 |
> | Step 4       | Cognitive Diagnosis                               | Diagnose based on response records to obtain ability vectors |
> | Step 5       | Adaptive Question Selection                       | Select questions adaptively from the question bank           |
> | Step 6       |                                                   | Repeat Steps 1–5 until the test is complete                  |
>
> **Report Generation**
>
> | Testing Step | Module Name         | Specific content                                             |
> | ------------ | ------------------- | ------------------------------------------------------------ |
> | Step 1       | Neural Architecture | Pass diagnosis model interaction records to a trained neural network to obtain a label (e.g., INTJ) |
> | Step 2       | Expert Analysis     | Combine expert analysis with neural network results for fine-tuning |
> | Step 3       | Diagnosis Report    | Output a diagnosis report based on the test results          |

---

> > ### Author Response · Authors · 2024-11-20
> >
> > > **Q3:** The paper discusses two key components of TestAgent: adaptive question selection and cognitive diagnosis. I think an ablation study would be helpful to evaluate the individual contributions of these components. It could also be interesting to see what happens if human experts take over one of these components, providing insight into human-machine collaboration.
> >
> > This is an interesting suggestion, but unfortunately, there is a strong interdependence between the components. The Adaptive Question Selection module relies on the results $\theta$ of the Cognitive Diagnosis model for question selection. If the question selection module is replaced with a human expert, the expert cannot select question based on $\theta$. Similarly, if the cognitive diagnosis model is replaced with an expert, the question selection module cannot select questions based on the textual information provided by the expert. The only way is for the expert to be responsible for both diagnosis and question selection, which is somewhat like an interview. This approach incurs significant testing costs because different domains require different experts to provide advice. Thank you for your suggestions, and we will consider hiring relevant experts to compare with our TestAgent in future work.
> >
> > > **Q4:** The appendix gives some examples of the tests, but it doesn’t show the full prompts sent to the agent. For an LLM-based system, the design of the prompts is crucial, as it defines the task and the agent's role. I suggest the authors provide more detail about how these prompts are structured to give a clearer picture of how the system operates.
> >
> > Thank you for your suggestion. Regarding prompts, we provide specific prompts in the paper, with detailed methods outlined in section E. This includes modules such as the Auto Feedback Mechanism, Anomaly Management, Question Response Generation, How to Transform Rigid Questions, and how to summarize the tester's response. These prompts can be directly utilized in specific large models. We also plan to open-source our code in future versions to provide more detailed explanations.
> >
> > We have also provided the specific details in the revised appendix, highlighted in blue text, please refer to Section B for more specific details. These include the fine-tuning process, detailed information about the datasets, classifier training, and the multidimensional evaluation experiment, among other details. If you have any questions, feel free to ask.

---

> ### Author Response · Authors · 2024-12-03
>
> The following is your correct review comments. This is from https://openreview.net/forum?id=UVaPEthRKx&noteId=9KJyDYslMg
>
> Summary:
> As an expert in large language models (LLMs), I must admit I’m not very familiar with cognitive diagnosis. So, my review will mostly focus on the LLM aspects. This paper introduces TestAgent, an adaptive testing agent based on LLMs, and honestly, the system has great potential. It conducts adaptive testing through interactive dialogues, which I think is a very cool idea. It helps solve some of the issues in traditional adaptive testing, like rigid question formats, noise in the data affecting accuracy, and overly simple output. By combining adaptive question selection and cognitive diagnosis, TestAgent dynamically selects personalized questions based on performance, identifies anomalies, and provides detailed diagnostic reports. What's even better is that it reduced the number of test questions by about 20% in several domains, such as psychological and educational testing, significantly improving testing efficiency.
>
> Soundness: 4: excellent
> Presentation: 3: good
> Contribution: 3: good
> Strengths:
> Originality: What impressed me the most is the combination of LLMs with adaptive testing. This is a fresh approach that breaks away from traditional, static testing, making the process much more flexible and interactive. The ability to generate personalized questions using LLMs really improves the overall testing experience.
>
> Quality: From the experimental results, TestAgent shows excellent performance in multiple fields. Reducing the number of test questions by 20% without sacrificing accuracy is incredibly valuable in real-world applications, saving time and maintaining reliability.
>
> Clarity: The paper is well-structured, and I particularly appreciated Figure 2, which clearly shows how the entire system works. Even readers who aren’t familiar with this area can easily understand the role of each module.
>
> Significance: In terms of practical applications, TestAgent has a lot of potential. Whether it's in education or mental health, personalized assessments and real-time feedback are crucial. A system that can dynamically adapt to user needs could have a big impact.
>
> Weaknesses:
> In the final step of the framework, expert analysis is still required. This feels a bit contradictory to the goal of improving automation. Since you're already using LLMs, why not go a step further and design an evaluation agent to replace or supplement the expert analysis? This could boost efficiency and reduce dependence on experts.
>
> While the framework diagram is clear, I think it could be made even more detailed. For example, showing the flow of data between different modules could make it easier to understand how the system operates, especially for readers unfamiliar with the field.
>
> The paper discusses two key components of TestAgent: adaptive question selection and cognitive diagnosis. I think an ablation study would be helpful to evaluate the individual contributions of these components. It could also be interesting to see what happens if human experts take over one of these components, providing insight into human-machine collaboration.
>
> The appendix gives some examples of the tests, but it doesn’t show the full prompts sent to the agent. For an LLM-based system, the design of the prompts is crucial, as it defines the task and the agent's role. I suggest the authors provide more detail about how these prompts are structured to give a clearer picture of how the system operates.
>
> Questions:
> In the final step of expert analysis, could an evaluation agent replace the expert? If so, how would this change the system’s automation and efficiency? Has this alternative been considered?
>
> Could you provide more detail about the prompts used to define the agent’s tasks? How do these prompts affect the agent’s behavior, especially in more complex testing scenarios?
>
> Have you considered conducting an ablation study to test the individual contributions of the adaptive question selection and cognitive diagnosis modules? How would the system perform if one of these components was replaced by expert intervention?
>
> Flag For Ethics Review: No ethics review needed.
> Details Of Ethics Concerns:
> No ethics review needed.
>
> Rating: 6: marginally above the acceptance threshold
> Confidence: 2: You are willing to defend your assessment, but it is quite likely that you did not understand the central parts of the submission or that you are unfamiliar with some pieces of related work. Math/other details were not carefully checked.
> Code Of Conduct: Yes

---

### Official Review · Reviewer_uQmS · 2024-11-03

**Soundness:** 3
**Presentation:** 3
**Contribution:** 2
**Rating:** 5
**Confidence:** 4

**Summary:**

The paper, presents TestAgent, a novel system that leverages large language models to improve the process of adaptive testing. Traditional adaptive testing methods, widely used in standardized assessments, face certain limitations, such as fixed-answer question formats, susceptibility to noisy data, and limited personalization in feedback. TestAgent addresses these challenges by employing a dynamic, conversational approach that allows the testing process to adjust in real-time based on each examinee’s responses. This design enables TestAgent to reduce the number of questions needed for precise assessment, achieving approximately 20% fewer questions while maintaining accuracy.

One of TestAgent's primary innovations lies in its ability to adapt questions interactively through natural language, making the testing experience more human-like and less mechanized. Additionally, TestAgent includes mechanisms for autonomous feedback and anomaly management, which detect inconsistencies in responses—such as guessing or ambiguous answers—and address them to enhance assessment reliability. The system also generates comprehensive diagnostic reports, providing examinees with detailed insights into their abilities and areas for improvement. In extensive testing across multiple domains, including educational, personality, and mental health assessments, TestAgent demonstrates significant performance gains in both accuracy and efficiency over traditional methods. By integrating LLMs into adaptive testing, TestAgent represents a significant step forward in creating personalized, conversational assessments that cater to individual user needs and provide a more interactive and engaging testing experience.

**Strengths:**

Originality: One of the paper’s most compelling strengths is its originality. By integrating large language models (LLMs) into the adaptive testing process, it introduces a novel application that creatively merges advanced language modeling with educational and psychological assessment. This approach innovates beyond traditional methods by offering a conversational, interactive experience in adaptive testing, thereby removing several limitations of rigid, fixed-question formats. The introduction of modules for autonomous feedback and anomaly management is particularly inventive, as it enhances the reliability of adaptive assessments through a system that actively manages ambiguous or inconsistent responses, a feature uncommon in prior adaptive testing approaches.

Quality: The research methodology is rigorous and well-executed, strengthening the quality of the paper’s contributions. The authors conducted extensive experiments on multiple datasets spanning educational, personality, and mental health domains, which demonstrate the model’s versatility and effectiveness. The choice of datasets and comparison with established baseline methods, such as random selection, FSI, KLI, and MAAT, ensure that the results are robust and credible. The use of metrics such as accuracy and area under the curve provides a clear quantitative evaluation of the model’s improvements, while the case studies highlight its practical applications.

Clarity: The paper is generally well-written and logically structured, making its technical content accessible to a broad audience. The problem formulation and objectives are clearly stated, and the contextual background effectively situates TestAgent within the larger field of adaptive testing. Figures and diagrams, such as the overall framework and case study examples, enhance the paper’s readability by visually illustrating the TestAgent’s processes and user interactions. Minor improvements in some sections, such as providing additional details on failure modes, could further enhance clarity, but overall, the writing style and organization serve the paper well.

Significance: The significance of this work is substantial, as it demonstrates how LLMs can enhance adaptive testing, an area of growing importance in personalized education, psychological evaluation, and beyond. By making assessments more interactive and adaptive, TestAgent aligns well with current trends toward personalized AI-driven experiences. The findings have meaningful implications for expanding the applications of LLMs beyond text generation, underscoring the broader potential of AI to enhance human-centered assessment tools. The results show a marked improvement in testing efficiency and accuracy, which could inspire further research and application in various fields, from educational technology to mental health diagnostics. This contribution is particularly relevant to the ICLR community, given its focus on advancements in machine learning and AI applications.

**Weaknesses:**

Although the paper demonstrates promising results, the experiments rely heavily on synthetic or simulated data to train and evaluate TestAgent. While this is a practical approach for initial testing, it limits the generalizability and applicability of the results to real-world scenarios, where user responses may be less predictable and more varied. Incorporating real-world data from actual assessments, such as real educational tests or personality assessments, could better validate the model’s effectiveness and adaptability. Additionally, discussing any potential discrepancies observed between synthetic and real data could strengthen the understanding of the model’s applicability and limitations in diverse user populations. The paper touches on the presence of errors, such as hallucinations and redundant answers, but it does not thoroughly analyze these issues or quantify their impact on the assessment quality. A deeper exploration of error types and their frequencies could reveal potential failure modes, helping to identify specific limitations or areas where the model struggles. For example, if hallucination errors frequently occur with specific question types, that insight could inform future improvements in the question generation or feedback mechanisms. Adding a robust error analysis, such as categorizing types of errors and discussing mitigation strategies, would provide a clearer understanding of the model’s robustness and improve the transparency of the results.

The paper lacks a discussion on the scalability of TestAgent, particularly when deployed at scale in real-world applications with a large number of test-takers. Given the computational requirements of LLMs, it’s likely that TestAgent requires significant processing power, which may limit its feasibility for institutions with resource constraints. Addressing this issue by either testing the system under constrained environments or proposing strategies to reduce computational load, such as distillation or pruning methods, could enhance its practical value. Discussing trade-offs between performance and efficiency in such settings would provide a more balanced view of the model’s strengths and limitations.

Although the impact statement briefly acknowledges fairness as a future area of research, it does not address potential biases that could arise from using LLMs in adaptive testing. Given that these models might reflect biases present in their training data, it is essential to examine if TestAgent’s question selection or feedback mechanisms favor certain groups or respond inconsistently across diverse demographic groups. Adding an initial analysis of bias or fairness, even at a high level, could demonstrate a commitment to addressing ethical concerns and provide actionable insights for future iterations. Testing TestAgent with users from various backgrounds and examining response patterns could serve as an important step toward ensuring fairness.

While the paper includes user evaluations of TestAgent’s performance, a more detailed comparative analysis of the user experience against traditional testing methods would add value. For example, analyzing user engagement, perceived transparency, and satisfaction with diagnostic feedback compared to conventional adaptive testing methods could highlight specific advantages and shortcomings. Additionally, tracking any learning curve or adaptation period required by users unfamiliar with conversational assessment formats could offer insights into usability and acceptance. Including more qualitative feedback from users, such as perceived accuracy or the helpfulness of the diagnostic report, would provide a holistic view of TestAgent’s effectiveness.

**Questions:**

Question: How does the model perform when evaluated on real-world data? Are there specific reasons for relying solely on synthetic data in the experiments?
Suggestion: Including experiments or case studies with real-world assessment data would strengthen the evidence base. If access to such data is restricted, discussing potential limitations that may arise when applying TestAgent to real users would be helpful.

Question: Could the authors provide a breakdown of the types of errors observed in more detail? For example, what percentage of errors were due to hallucinations, misinterpretations, or other issues?
Suggestion: A thorough analysis of error types, including quantification and discussion of the impact on assessment quality, would improve transparency and demonstrate robustness. Could specific adjustments in the feedback or anomaly modules mitigate these errors?


Question: Have the authors considered potential biases that might arise in the question selection or response evaluation mechanisms? Are there steps in place to identify or mitigate these biases?
Suggestion: Adding a preliminary analysis of fairness or bias, even if high-level, could address concerns about the model’s consistent behavior across diverse demographic groups. It would also provide insight into the challenges and plans for ensuring equity in assessments.


Question: Can the authors elaborate on the computational requirements for running TestAgent, especially in resource-constrained environments?
Suggestion: A discussion on scalability, including potential strategies to reduce computational load provide practical insights for broader applicability.



Question: How does TestAgent compare with traditional adaptive testing methods in terms of user engagement and satisfaction with the testing process?
Suggestion: Incorporating a more detailed comparative analysis on user experience, perhaps by analyzing user satisfaction or perceived accuracy against conventional methods, could highlight TestAgent’s unique benefits. Any qualitative feedback on the diagnostic reports would also be informative.



Question: How effectively does the Autonomous Feedback Mechanism handle ambiguous responses in practice, and are there thresholds for when a response is deemed too ambiguous?
Suggestion: Providing examples of ambiguous responses and explaining how the feedback mechanism resolves them would clarify its practical effectiveness. If there are predefined thresholds for ambiguity, detailing these would improve understanding of the system’s robustness.

---

> ### Author Response · Authors · 2024-11-15
>
> Thank you for your thoughtful review and the recognition of our paper. We will address each of your questions below.
>
> > **Q1:** Question: How does the model perform when evaluated on real-world data? Are there specific reasons for relying solely on synthetic data in the experiments?
>
> Although our MBTI and SCL datasets are synthetic, the MATH dataset is a real dataset, obtained from a private company. Detailed information about the dataset is shown in the table. For synthetic datasets, if there is a corresponding real-world dataset available, using the real dataset is undoubtedly the best option. However, unfortunately, it is very difficult to access interaction data between questions and test-takers for every domain, as this information is relatively sensitive, especially in areas like mental health and personality testing. Our framework aims to propose a universal method for data construction and cognitive diagnosis model training that can be applied across various domains. We hope to provide a concrete solution for the majority of fields, and therefore, the validation of synthetic datasets is crucial.
>
> > **Q2:** Question: Could the authors provide a breakdown of the types of errors observed in more detail? For example, what percentage of errors were due to hallucinations, misinterpretations, or other issues?
>
> We have already provided the types of errors and their specific proportions in the experiments, along with some examples in the appendix(Line 506). The errors mentioned in the paper occur only in rare cases. We have collected data on the errors from the above-mentioned experiments. Despite using modules like Anomaly Management and Autonomous Feedback System to reject unreasonable content generated by the large model, and incorporating the Expert Analysis module for fine-tuning to reduce the model’s hallucinations, it is not possible to completely eliminate these issues. The occurrence of such errors is primarily due to the inherent limitations of the large model and the errors introduced by randomness. In future work, we will provide a more detailed discussion of the specific types of errors.
>
> > **Q3:** The paper lacks a discussion on the scalability of TestAgent, particularly when deployed at scale in real-world applications with a large number of test-takers.
>
> Our paper focuses on proposing a universal testing framework that can transform existing tests into adaptive and intelligent testing formats. Regarding scalability, TestAgent has already transformed traditional rigid tests into role-playing agent-based tests. One obvious scalability feature is the use of prompt engineering, where different prompts are tailored to individuals of different ages and roles, ensuring the best testing outcomes. Moreover, existing question-generation plugins and other adaptive question-selection algorithms can also be integrated into TestAgent. The reason we have not provided a more extensive discussion on scalability is that it would make the discussion quite complex. Each component in Figure 2 can be extended as needed, so we only provide the basic framework, with modifications to be made on top of the corresponding components in Figure 2.
>
> > **Q4:** Question: How effectively does the Autonomous Feedback Mechanism handle ambiguous responses in practice, and are there thresholds for when a response is deemed too ambiguous?
>
> For label determination, the paper provides a detailed explanation (Line494). For cases where label determination is difficult, we offer three rules to assist the large model in label assignment. These rules are based on three perspectives: domain relevance (Line187), response alignment (Line192), and logical coherence (Line197), to avoid such situations. Once the Autonomous Feedback Mechanism determines a failure, the corresponding measurement step is re-executed.
>
> Our paper primarily focuses on proposing a novel testing framework that transforms existing tests into adaptive and intelligent testing formats. It aims to break through the bottlenecks of traditional testing and solve existing problems. This is the core of our paper. We appreciate the reviewer's suggestions, but some of the points raised are not relevant to this paper. For instance, the suggestions regarding **large model performance** , **user experience analysis**, and **high-level fairness** issues fall outside the scope of our work, as these are not directly related to the focus of our study. The suggestions you mentioned belong to other independent domains and are not related to our research. If you have any further questions, feel free to discuss them with us.

---

### Official Review · Reviewer_2J8D · 2024-11-03

**Soundness:** 2
**Presentation:** 1
**Contribution:** 2
**Rating:** 3
**Confidence:** 4

**Summary:**

This work introduces TestAgent, an LLM-based adaptive testing agent. TestAgent has dynamic question selection capabilities, an autonomous feedback mechanism and anomaly management module, and the ability to generate detailed diagnosis reports. The authors use student data from MATH education, personality tests, and mental health testing. As part of the results, the authors present simulation-based evaluation results, a case study with Test Agent, and a human study with volunteers to interact with TestAgent.

**Strengths:**

- The work attempts to tackle an important problem of human measurement. It seems like a good idea to leverage LLM-based agents for this purpose.
- The work tries to study their proposed systems through a series of simulated and real-world experiments.

**Weaknesses:**

- Overall, the writing and organization of the paper are hard to follow and thus make it hard to provide a nuanced assessment of the submission. There are also many typos throughout the work—please proofread carefully. The remaining comments will talk more about suggestions for improving the presentation of methodology as well as recommendations for evaluation clarity.
- What exactly is being measured through human assessments is unclear throughout Section 2, i.e. what is exactly captured by $\theta$ and what are concrete examples as defined by the datasets considered in the experimental section?
- It was challenging to understand how each component of the TestAgent framework depicted in Figure 2 was studied in the evaluation. It seemed like there were many moving parts and was unclear whether the entire framework was being evaluated or just components. Ideally, the evaluation conducted in Section 3 would clearly ablate different individual components.
- There was a general lack of details in the results in terms of methodology. In Section 3.1, the authors say “We fine-tuned the ChatGLM2- 6B (GLM et al., 2024) series using comprehensive expert diagnosis reports and synthetic datasets as fine-tuning data.” This type of language glosses over a lot of the details about what the fine-tuning process looked like, what exact datasets were used, and how synthetic datasets were generated. Another example is in L264, where the authors say “we train a classifier” without further details.
- The experimental setup could also be described in much more detail. The authors describe using student data in simulation experiments. How much data were there in each dataset? Where was this data obtained from? These datasets seem to be sourced from very different domains. Section 3.7 needs to be significantly elaborated on. Were the volunteers trained to perform these types of assessments? What kinds of instructions were they given? What version of TestAgent was used? Please provide more of these details in the main text and Appendix.
- There seem to be some issues regarding the claims of effectiveness. In Table 1, the authors say that “the bold text indicates statistically significant superiority (p-value ≤ 0.01) over the best baseline”. How is this possible given some of the results in the table? For example, in Table 1 (a) MAAT has an AUC@50 of 65.12+/-1.48 and TestAgent+KLI an AUC@50 of 65.12+/-1.48, and yet only the latter is bolded. Please provide more detail on what statistical tests were conducted.

**Questions:**

Please address the specific questions raised in the weaknesses section.

---

> ### Author Response · Authors · 2024-11-15
>
> We would like to express our sincere gratitude for taking the time to review our paper. Your insights are  valuable to us. Below, we provide our responses to the points you raised."
>
> > **Q1:** What exactly is being measured through human assessments is unclear throughout Section 2, i.e. what is exactly captured by θ and what are concrete examples as defined by the datasets considered in the experimental section?
>
> Psychometrics typically represents a test-taker's ability as a vector $ \theta $. In Item Response Theory (IRT), $ \theta $ also represents the test-taker's ability, where a higher value indicates stronger proficiency. This is explained in more detail in our article. For example, in the MATH dataset, which is a student math test, $ \theta $ measures the student's ability level. The higher the value of $ \theta $, the stronger the ability, and the higher the probability of answering questions correctly. In the MBTI dataset, $ \theta $ is represented as a four-dimensional vector. For instance, in the (I/E) dimension, the larger the value of $ \theta $ in this dimension, the more extroverted (E) the individual is.
>
> > **Q2:** It was challenging to understand how each component of the TestAgent framework depicted in Figure 2 was studied in the evaluation. It seemed like there were many moving parts and was unclear whether the entire framework was being evaluated or just components.
>
> It seems there has been some misunderstanding. In Section 2, we generally introduced the purpose of each component, and the components work in collaboration, not as isolated moving parts. The representation in Figure 2 is intended to clearly show the role of each module, rather than suggesting that each module functions independently. Our paper does not analyze from the perspective of individual components; rather, it focuses on the entire TestAgent framework. Therefore, in the final experimental section, it is not possible to evaluate the components separately, as the evaluation requires their joint collaboration to achieve the intelligent testing of TestAgent.
>
> The reviewer expressed interest in some of the training details of our paper, and we provide the following explanation.
>
> **Ability Classifier Training**
>
> Cognitive diagnostic models provide a vector $\theta$ as the diagnostic result; however, this is not interpretable. For the vector $\theta$, the MBTI test includes four dimensions: (I/E), (N/S), (T/F), and (J/P). Therefore, we train a classifier where the input is the diagnostic model's $\theta$, and the output is a four-dimensional vector corresponding to these four dimensions, thus transforming the abstract diagnostic number into features.
>
> In specific terms, for a personality classification data labeled as $ Y_{label} $, cognitive diagnostics provide a diagnostic result $\theta$ based on response to questions. Let $ f $ be a mapping function that can map personality classifications to a 0-1 vector, for example, $ f('ENFJ') = [1, 0, 1, 0] $. Let $ g $ be the classifier we aim to train. The loss function can then be written as:
>
> $L(\theta) = CrossEntropy Loss(  g(\theta) , f(Y_{label} ) )$. With this, the classifier training can be implemented.
>
> **Performance Test Details**
>
> We repeated the experiment several times for each baseline. We used t-tests to assess the performance differences between different methods. Specifically, for each pairwise comparison of methods, we conducted independent samples t-tests under the assumption of equal variances. We tested whether the differences in the ACC and AUC metrics were statistically significant. For results with a p-value ≤ 0.01, we considered the difference to be statistically significant and marked it in bold in the table. The bolded text you mentioned is due to the significant advantage of TestAgent+KLI over the other baseline models. Although the AUC@50 values are identical between TestAgent+KLI and MAAT, the best result for TestAgent+KLI is 67.00, while the best result for MAAT is 66.60. Therefore, we have highlighted this difference using bold text.

---

> > ### Author Response · Authors · 2024-11-15
> >
> > **Fine-tune Details:**
> >
> > In this study, the fine-tuning process is based on the pre-trained ChatGLM model, aiming to customize the model for the specific personality diagnostic task to improve its performance in handling MBTI personality analysis tasks. To achieve this, we perform fine-tuning using LoRA (Low-Rank Adaptation) technology through the torchkeras framework.
> >
> > **Data Processing:**
> >
> > The fine-tuning data is divided into three parts: instructions, character labels, and expert reports.
> >
> > - The instruction is a simple prompt, formatted as follows:
> >
> >   "Based on personality test classification and relevant dialogues, analyze the character traits and provide the corresponding diagnostic report."
> >
> > - Character labels include the labels obtained through the ability classifier training mentioned earlier.
> >
> > - Expert reports are the personality diagnostic reports provided by the official MBTI website for the 16 personality types.
> >
> > Each piece of fine-tuning data consists of an input formed by combining the instruction and the character label, and the output is the diagnostic report suggestion, which corresponds to the expert's diagnostic report. Thus, the construction of fine-tuning data is completed.
> >
> > **Parameter Settings:**
> >
> > In this work, several hyperparameters are carefully chosen for fine-tuning the model. The maximum sequence length is set to 1024 tokens, ensuring that input sequences longer than this are truncated.
> >
> > For the Low-Rank Adaptation (LoRA) method, three key parameters are used:
> >
> > - The rank $ r $ is set to 8, which controls the size of the low-rank matrices.
> > - The scaling factor $ \alpha $ is set to 32, which adjusts the influence of the low-rank adaptation.
> > - The dropout rate $ p $ is set to 0.1, which applies a 10% dropout during training to help with regularization.
> >
> > Training-related hyperparameters include:
> >
> > - A batch size of 8,
> > - A learning rate of $ 2 \times 10^{-6} $,
> > - A total of 10 training epochs.
> >   Additionally, early stopping is applied with a patience of 2 epochs, meaning that training stops if the validation loss does not improve over 3 consecutive epochs.
> >
> > Finally, mixed precision training is employed with a setting of 'fp16' to improve computational efficiency, and when saving the model, the maximum shard size is set to 1GB, ensuring that the model is saved in manageable chunks for later use.
> >
> > **Dataset Information**
> >
> > Here we provide specific information for each dataset, along with concrete examples. The table displays the number of students, the number of questions, and the count of interaction responses for each dataset.
> >
> > | **DATASETS** | **Number of Testers** | **Number of Questions** | **Number of Responses** |
> > | ------------ | --------------------- | ----------------------- | ----------------------- |
> > | MBTI         | 1000                  | 60                      | 60000                   |
> > | SCL          | 500                   | 90                      | 45000                   |
> > | MATH         | 1940                  | 1485                    | 61860                   |
> >
> > Below are some specific question contents.
> >
> > - **MBTI**: *Your personal working style leans more towards spontaneous bursts of energy rather than systematic and sustained effort.*
> > - **SCL**: *Feeling a decrease in energy and a slowing down of activities.*
> > - **MATH**: *For a cylinder with a base radius of 1 and a height of 1, the surface area of the cylinder is.*

---

> > > ### Author Response · Authors · 2024-11-15
> > >
> > > ### Multidimensional Evaluation Details
> > >
> > > The multidimensional evaluation experiment involves 50 volunteers from different fields, who score on four dimensions: accuracy, fluency, speed, and interaction.
> > >
> > > - **Accuracy** refers to how well the volunteer's results align with their actual situation and whether the final diagnostic recommendations are accurate.
> > > - **Fluency** represents the smoothness of the test.
> > > - **Speed** refers to the time taken to complete the test.
> > > - **Interaction** measures the level of interactivity in the testing experience.
> > >
> > > However, human labeling can be subject to bias, which is inevitable. To reduce this bias, we have selected volunteers of varying gender, age, and educational background for the test.
> > >
> > > | **Category**        | **Percentage** |
> > > | ------------------- | -------------- |
> > > | **Gender**          |                |
> > > | Male                | 62%            |
> > > | Female              | 38%            |
> > > | **Age**             |                |
> > > | 10-18 years old     | 10%            |
> > > | 18-30 years old     | 46%            |
> > > | 30-40 years old     | 20%            |
> > > | 40-60 years old     | 18%            |
> > > | 60-70 years old     | 6%             |
> > > | **Education Level** |                |
> > > | College degree      | 46%            |
> > > | No college degree   | 54%            |
> > >
> > > We performed significance testing. We conducted hypothesis testing across different dimensions to eliminate bias in human annotations. The specific data is as follows:
> > >
> > > **Gender: Independent Samples t-test**
> > >
> > > **Null Hypothesis (H₀):** There is no significant difference in the mean scores between males and females on a given dimension. That is, the mean scores of males and females are equal.
> > >
> > > **Alternative Hypothesis (H₁):** There is a significant difference in the mean scores between males and females on the given dimension. That is, the mean scores of males and females are different.
> > >
> > > | **Dimension** | **t-statistic** | **p-value** |
> > > | ------------- | --------------- | ----------- |
> > > | Accuracy      | -1.34           | 0.1805      |
> > > | Fluency       | -1.49           | 0.1372      |
> > > | Speed         | 1.05            | 0.2945      |
> > > | Experience    | 0.95            | 0.3416      |
> > >
> > > Cannot reject the null hypothesis.
> > >
> > > **Age: One-Way ANOVA**
> > >
> > > **Null Hypothesis (H₀):** There is no significant difference in the mean scores between the different age groups. That is, the scores of different age groups are similar.
> > >
> > > **Alternative Hypothesis (H₁):** At least one age group has a mean score that is different from the others. That is, there is a significant difference in scores between age groups.
> > >
> > > | **Dimension** | **F-statistic** | **p-value** |
> > > | ------------- | --------------- | ----------- |
> > > | Accuracy      | 1.0218          | 0.3971      |
> > > | Fluency       | 2.1243          | 0.079       |
> > > | Speed         | 2.0162          | 0.0936      |
> > > | Experience    | 1.3827          | 0.2413      |
> > >
> > > Cannot reject the null hypothesis.
> > >
> > > **Education Level:**
> > >
> > > **Null Hypothesis (H₀):** There is no significant difference in the mean scores between testers who have attended college and those who have not on a given dimension.
> > >
> > > **Alternative Hypothesis (H₁):** There is a significant difference in the mean scores between testers who have attended college and those who have not on the given dimension.
> > >
> > > | **Dimension** | **t-statistic** | **p-value** |
> > > | ------------- | --------------- | ----------- |
> > > | Accuracy      | -1.29           | 0.1984      |
> > > | Fluency       | 1.06            | 0.2867      |
> > > | Speed         | 1.26            | 0.2084      |
> > > | Experience    | -0.29           | 0.7653      |
> > >
> > > Cannot reject the null hypothesis.
> > >
> > > **Independent Samples t-test**
> > >
> > > We use the **paired t-test** to compare the score differences between traditional tests and TestAgent across each dimension.
> > >
> > > **Null Hypothesis (H₀):** There is no significant difference between traditional tests and TestAgent on a given dimension.
> > >
> > > **Alternative Hypothesis (H₁):** TestAgent outperforms traditional tests on the given dimension. Statistically, if the p-value is less than 0.05, the novel test on this dimension is considered significantly better than the traditional test.
> > >
> > > | **Dimension** | **t-statistic** | **p-value** |
> > > | ------------- | --------------- | ----------- |
> > > | Accuracy      | -2.56           | 0.01188     |
> > > | Fluency       | -6.53           | 2.80e-09    |
> > > | Speed         | -6.09           | 2.11e-08    |
> > > | Experience    | -6.46           | 3.87e-09    |
> > >
> > > The experiment shows that the null hypothesis is rejected.
> > >
> > > We have also provided the specific details in the revised appendix, highlighted in blue text, please refer to Section B for more specific details. These include the fine-tuning process, detailed information about the datasets, classifier training, and the multidimensional evaluation experiment, among other details. We hope this can address the reviewer's concerns. Please feel free to ask if you have any questions.

---

### Official Review · Reviewer_GeR3 · 2024-11-05

**Soundness:** 3
**Presentation:** 2
**Contribution:** 2
**Rating:** 5
**Confidence:** 4

**Summary:**

The paper describes an LLM based agent system designed to perform human assessments in an optimal manner. Optimal meaning as few questions as necessary to achieve an accurate assessment. The assessment is applied using chat interaction with the user. The system dynamically selects questions, represents the users performance (cog. diagnosis), and attempts to identify and correct for various anomalous answers via open ended elaboration. Report generation is also covered.

**Strengths:**

The idea here is good and the solution is comprehensive. At a high level the authors are proposing to use LLMs to perform human assessment vs. using existing rigid approaches. This intuitively makes sense, but of course is not straight forward due to the inclusion of a potentially unreliable model.

Representing the users performance based on previous answers, and using this representation to implement anomaly detection is an interesting concept. In general the anomaly detection component is compelling.

The focus on various types of evaluation is helpful, particularly the identification of error modes.

**Weaknesses:**

The Universal Data Infrastructure is not well defined, I came away not really understanding this component. Please revise the description with a focus on clarity. Specifically cognitive diagnosis training.

The major weakness here is that using an LLM introduces a new problem, namely LLM hallucinations leading to misrepresentation of results.  The authors bring up this up in section 3.8. It would be worthwhile for the authors to discuss the trade off between the introduction of hallucinations and the issues of traditional testing that originally spurred the work.

The multidimensional evaluation omits many details necessary to determine it validity i.e. how is accuracy determined? Controls for ordering and bias? Significance tests.

**Questions:**

What kind of model is the cognitive diagnosis component? What data is it trained on and how is it applied? getting very concrete here will help.

How was accuracy determined in the multidimensional evaluation?

---

> ### Author Response · Authors · 2024-11-15
>
> Thank you for taking the time to read this paper and provide your valuable suggestions. Your feedback is greatly appreciated. Below, we will address the concerns you raised.
>
> > **Q1：** The Universal Data Infrastructure is not well defined, I came away not really understanding this component. Please revise the description with a focus on clarity. Specifically cognitive diagnosis training.
>
> The Universal Data Infrastructure consists of three main steps. First, Domain Verification is used to identify the testing domain. For example, for a mathematics ability test, the testing dimension is determined.
>
> Next, in the Data Integration module, GPT is used to simulate students and generate response data. The data pair consists of Student - Question - Label. These response records are then passed on to the Cognitive Diagnosis Training module.
>
> The purpose of Cognitive Diagnosis is to analyze the current student ability $\theta$ based on the response data. Taking the 3PL-IRT as an example, the model is expressed as:
> $P_i(\theta) = c_i + (1 - c_i) \cdot \frac{1}{1 + e^{-(a_i(\theta - b_i))}}$
>
> Each question $ i $ has characteristics such as discrimination $ a $, difficulty $ b $, and guessing parameter $ c $. The goal of our Universal Data Infrastructure is to abstract the features $ a $, $ b $, and $ c $ for each question across various types of tests. This allows the cognitive diagnosis model to analyze the data effectively.
>
> Thus, Cognitive Diagnosis Training is the process of training the features $ a $, $ b $, and $ c $. The loss function is as follows:
> $L(\theta,Y) = \sum_{i} -\left(y_i \log( P_i(\theta)) + (1 - y_i) \log(1 - P_i(\theta)) \right)$
>
> This loss function calculates the cross-entropy loss between the true labels $ y $ and the cognitive diagnosis $ P(\theta) $, which is then used for training. This is the process of the Universal Data Infrastructure.
> > **Q2:** The major weakness here is that using an LLM introduces a new problem, namely LLM hallucinations leading to misrepresentation of results. The authors bring up this up in section 3.8. It would be worthwhile for the authors to discuss the trade off between the introduction of hallucinations and the issues of traditional testing that originally spurred the work.
>
> Traditional testing is relatively rigid. Based on the capabilities of large models, we have developed the TestAgent framework to introduce innovation in testing formats, breaking free from the limitations of traditional tests. However, the issue of hallucinations in large models is unavoidable. We have tried to mitigate this effect by using modules such as Anomaly Management and Autonomous Feedback System, which help reject unreasonable content generated by the large model. Additionally, we incorporated an Expert Analysis module for fine-tuning to reduce the hallucinations. However, the hallucination issue cannot be completely eliminated. In future work, we plan to integrate expert evaluations into the framework to balance the hallucination problem, especially for high-stakes exams. The innovation of this paper lies in proposing an innovative testing framework that addresses issues like randomness and rigidity in traditional tests. We deeply appreciate your suggestions, which will be discussed in more detail in future work.

---

> > ### Author Response · Authors · 2024-11-15
> >
> > > **Q3：** The multidimensional evaluation omits many details necessary to determine it validity i.e. how is accuracy determined? Controls for ordering and bias? Significance tests.
> >
> > The multidimensional evaluation experiment involves 50 volunteers from different fields, who score on four dimensions: accuracy, fluency, speed, and interaction.
> >
> > - **Accuracy** refers to how well the volunteer's results align with their actual situation and whether the final diagnostic recommendations are accurate.
> > - **Fluency** represents the smoothness of the test.
> > - **Speed** refers to the time taken to complete the test.
> > - **Interaction** measures the level of interactivity in the testing experience.
> >
> > However, human labeling can be subject to bias, which is inevitable. To reduce this bias, we have selected volunteers of varying gender, age, and educational background .
> >
> > | **Category**        | **Percentage** |
> > | ------------------- | -------------- |
> > | **Gender**          |                |
> > | Male                | 62%            |
> > | Female              | 38%            |
> > | **Age**             |                |
> > | 10-18 years old     | 10%            |
> > | 18-30 years old     | 46%            |
> > | 30-40 years old     | 20%            |
> > | 40-60 years old     | 18%            |
> > | 60-70 years old     | 6%             |
> > | **Education Level** |                |
> > | College degree      | 46%            |
> > | No college degree   | 54%            |
> >
> > We conducted hypothesis testing across different dimensions to eliminate bias in human annotations. The specific data is as follows:
> >
> > **Gender: Independent Samples t-test**
> >
> > **Null Hypothesis (H₀):** There is no significant difference in the mean scores between males and females on a given dimension. That is, the mean scores of males and females are equal.
> >
> > **Alternative Hypothesis (H₁):** There is a significant difference in the mean scores between males and females on the given dimension. That is, the mean scores of males and females are different.
> >
> > | **Dimension** | **t-statistic** | **p-value** |
> > | ------------- | --------------- | ----------- |
> > | Accuracy      | -1.34           | 0.1805      |
> > | Fluency       | -1.49           | 0.1372      |
> > | Speed         | 1.05            | 0.2945      |
> > | Experience    | 0.95            | 0.3416      |
> >
> > Cannot reject the null hypothesis.
> >
> > **Age: One-Way ANOVA**
> >
> > **Null Hypothesis (H₀):** There is no significant difference in the mean scores between the different age groups. That is, the scores of different age groups are similar.
> >
> > **Alternative Hypothesis (H₁):** At least one age group has a mean score that is different from the others. That is, there is a significant difference in scores between age groups.
> >
> > | **Dimension** | **F-statistic** | **p-value** |
> > | ------------- | --------------- | ----------- |
> > | Accuracy      | 1.0218          | 0.3971      |
> > | Fluency       | 2.1243          | 0.079       |
> > | Speed         | 2.0162          | 0.0936      |
> > | Experience    | 1.3827          | 0.2413      |
> >
> > Cannot reject the null hypothesis.
> >
> > **Education Level:**
> >
> > **Null Hypothesis (H₀):** There is no significant difference in the mean scores between testers who have attended college and those who have not on a given dimension.
> >
> > **Alternative Hypothesis (H₁):** There is a significant difference in the mean scores between testers who have attended college and those who have not on the given dimension.
> >
> > | **Dimension** | **t-statistic** | **p-value** |
> > | ------------- | --------------- | ----------- |
> > | Accuracy      | -1.29           | 0.1984      |
> > | Fluency       | 1.06            | 0.2867      |
> > | Speed         | 1.26            | 0.2084      |
> > | Experience    | -0.29           | 0.7653      |
> >
> > Cannot reject the null hypothesis.
> >
> > **Independent Samples t-test**
> >
> > We use the **paired t-test** to compare the score differences between traditional tests and TestAgent across each dimension.
> >
> > **Null Hypothesis (H₀):** There is no significant difference between traditional tests and TestAgent on a given dimension.
> >
> > **Alternative Hypothesis (H₁):** TestAgent outperforms traditional tests on the given dimension. Statistically, if the p-value is less than 0.05, the novel test on this dimension is considered significantly better than the traditional test.
> >
> > | **Dimension** | **t-statistic** | **p-value** |
> > | ------------- | --------------- | ----------- |
> > | Accuracy      | -2.56           | 0.01188     |
> > | Fluency       | -6.53           | 2.80e-09    |
> > | Speed         | -6.09           | 2.11e-08    |
> > | Experience    | -6.46           | 3.87e-09    |
> >
> > The experiment shows that the null hypothesis is rejected, indicating that TestAgent significantly outperforms traditional tests in terms of accuracy, fluency, test speed, and interaction experience.
> >
> > We have also provided the specific details in the revised appendix, highlighted in blue text, please refer to Section B for more specific details. Please feel free to ask if you have any questions.

---

> > > ### Comment · Reviewer_GeR3 · 2024-11-26
> > >
> > > Regarding Q3. I appreciate the response, but its still unclear to me how accuracy was determined?

---

> > > > ### Author Response · Authors · 2024-11-27
> > > >
> > > > Thank you for your reply. In response to the concerns you still have, we have provided further clarification.
> > > >
> > > > > How is the dimension determined? How is it represented, and how is it used?
> > > >
> > > > Every test has its own dimensions. For example, in a mathematics ability test, the question judgment is either correct or incorrect, so it is considered to have 2 dimensions. For MBTI tests, such as the 16 personality test, the official options are divided into 7 parts, representing 0: Completely Disagree, 1: Strongly Disagree, 2: Mildly Disagree, 3: Neutral, 4: Mildly Agree, 5: Strongly Agree, 6: Completely Agree, thus the test has 7 dimensions.
> > > >
> > > > >GPT is used to simulate students What does the prompt look like? How much data is generated? Can we see some examples?
> > > >
> > > > Here we provide specific information for each dataset, along with concrete examples. The table displays the number of students, the number of questions, and the count of interaction responses for each dataset.
> > > >
> > > > | **DATASETS** | **Number of Testers** | **Number of Questions** | **Number of Responses** |
> > > > | ------------ | --------------------- | ----------------------- | ----------------------- |
> > > > | MBTI         | 1000                  | 60                      | 60000                   |
> > > > | SCL          | 500                   | 90                      | 45000                   |
> > > > | MATH         | 1940                  | 1485                    | 61860                   |
> > > >
> > > > Below are some specific question contents.
> > > >
> > > > - **MBTI**: *Your personal working style leans more towards spontaneous bursts of energy rather than systematic and sustained effort.*
> > > > - **SCL**: *Feeling a decrease in energy and a slowing down of activities.*
> > > > - **MATH**: *For a cylinder with a base radius of 1 and a height of 1, the surface area of the cylinder is.*
> > > >
> > > > We provided the specific data generation prompt in Table 11, and the generated data example from the prompt is as follows:
> > > >
> > > > ```
> > > > {
> > > > "Question1": {"Answer": 0, "Response": "I feel that making new friends really exhausts me, so I don't want to make new friends."},
> > > > "Question2": {"Answer": 3, "Response": "I always spend a lot of time exploring various topics that interest me because I believe that the depth and breadth of knowledge are important to me."},
> > > > ...
> > > > "Question60": {"Answer": 1, "Response": "I believe things will develop logically rather than in my favor."}
> > > > }
> > > > ```
> > > >
> > > > > Where does the a, b and c variables come from?
> > > >
> > > > a, b, and c represent difficulty, discrimination, and guessing parameters. These are variables inherent in the Item Response Theory (3PL-IRT) model and are used to better fit the test [1]. Our goal is to train the parameters a, b, and c for each test question. We treat a, b, and c as trainable parameters, which are randomly initialized and then updated using generated data through gradient descent. It is based on the training approach mentioned above in Q1.
> > > >
> > > > > How was accuracy determined in the multidimensional evaluation?
> > > >
> > > > Accuracy is derived from the subjective feedback of volunteers. To reduce the influence of participant subjectivity, we provide standardized criteria for accuracy to guide the volunteers. The accuracy evaluation is divided into two aspects: personality label accuracy and diagnosis report accuracy. The rating scale is from 1 to 5, with the following definitions:
> > > >
> > > > - 1: The personality label is completely incorrect, and the diagnosis report is completely inconsistent.
> > > > - 2: The personality label is partially correct, and the diagnosis report is partially inconsistent.
> > > > - 3: The personality label is partially correct, and the diagnosis report is mostly consistent.
> > > > - 4: The personality label is accurate, and the diagnosis report is mostly consistent.
> > > > - 5: The personality label is accurate, and the diagnosis report is completely consistent.
> > > >
> > > > [1] Woods C M. Ramsay-curve item response theory for the three-parameter logistic item response model[J]. Applied Psychological Measurement, 2008, 32(6): 447-465.

---

> > ### Comment · Reviewer_GeR3 · 2024-11-26
> >
> > Thank you for addressing my questions.
> > The explanation helped somewhat but unfortunately Im still not convinced re. the Universal Data Infrastructure.
> >
> > > First, Domain Verification is used to identify the testing domain. For example, for a mathematics ability test, the testing dimension is determined.
> >
> > How is the dimension determined? How is it represented, and how is it used?
> >
> > > GPT is used to simulate students
> > What does the prompt look like? How much data is generated? Can we see some examples?
> >
> > Where does the a, b and c variables come from?

---

### Author Response · Authors · 2024-11-27
**Global Response**

We are very grateful to each reviewer for their careful evaluation and valuable comments on the manuscript. Their thoughtful reviews provide valuable insights for improving the manuscript. The positive feedback from the reviewers in various aspects has greatly encouraged us.

- **Contribution**:
  "The idea here is good and the solution is comprehensive." (GeR3)
  "This approach innovates beyond traditional methods by offering a conversational, interactive experience in adaptive testing." (uQmS)
  "The work attempts to tackle an important problem of human measurement." (2J8D)
  "TestAgent represents a significant step forward in creating personalized, conversational assessments." (uQmS)
- **Presentation**:
  "The paper is well-structured, and I particularly appreciated Figure 2, which clearly shows how the entire system works." (mcjy)
  "The paper is generally well-written and logically structured, making its technical content accessible to a broad audience." (uQmS)
  "The paper is generally well-written, and the contextual background effectively situates TestAgent within the larger field of adaptive testing." (uQmS)
- **Method**:
  "Representing the users' performance based on previous answers, and using this representation to implement anomaly detection is an interesting concept. In general, the anomaly detection component is compelling." (GeR3)
  "The introduction of modules for autonomous feedback and anomaly management is particularly inventive." (uQmS)
  "What impressed me the most is the combination of LLMs with adaptive testing. This is a fresh approach that breaks away from traditional, static testing, making the process much more flexible and interactive." (mcjy)
- **Experimental Results**:
  "Reducing the number of test questions by 20% without sacrificing accuracy is incredibly valuable in real-world applications, saving time and maintaining reliability." (mcjy)
  "The research methodology is rigorous and well-executed, strengthening the quality of the paper’s contributions." (uQmS)
  "The focus on various types of evaluation is helpful, particularly the identification of error modes." (GeR3)

Our work is dedicated to advancing intelligent testing by integrating large language model (LLM) technology with adaptive testing practices. We leverage flexible, adaptive intelligent testing agents to provide a more personalized, efficient, and innovative testing experience. Our goal is to introduce the concept and technology of combining large pre-trained models with intelligent testing to the ICLR community. This not only brings new research directions and ways of thinking to the community but also offers its members broader research fields and collaboration opportunities.

With regard to the questions raised by each reviewer, we carefully considered each point and responded in detail in each rebuttal pane.

---

### Note · Authors · 2024-12-09

I have read and agree with the venue's withdrawal policy on behalf of myself and my co-authors.